# Whole-genome sequencing reveals a possible molecular basis of sex determination in the dioecious wild yam *Dioscorea tokoro*

Aoi Kudoh[1], Satoshi Natsume[2], Yu Sugihara[3], Hiroaki Kato[1,4], Akira Abe[2],
Kaori Oikawa[2], Motoki Shimizu[2], Kazue Itoh[2], Mai Tsujimura[5], Yoshitaka Takano[6],
Toshiyuki Sakai[1], Hiroaki Adachi[1,7], Atsushi Ohta[1], Mina Ohtsu[7], Takuma Ishizaki[8],
Toru Terachi[9], Hideki Innan[10], Ryohei Terauchi[1,2]*

1 Laboratory of Crop Evolution, Graduate School of Agriculture, Kyoto University, Kyoto, Japan, 2 Iwate Biotechnology Research Center, Kitakami, Iwate, Japan, 3 The Sainsbury Laboratory, University of East Anglia, Norwich Research Park, Norwich, United Kingdom, 4 JST-PRESTO, Kawaguchi, Saitama, Japan, 5 Department of Plant Life Science, Faculty of Agriculture, Ryukoku University, Otsu, Shiga, Japan, 6 Laboratory of Plant Pathology, Graduate School of Agriculture, Kyoto University, Kyoto, Japan, 7 Laboratory of Plant Immunity, Graduate School of Life Science, Hokkaido University, Hokkaido, Japan, 8 Japan International Research Center for Agricultural Sciences (JIRCAS), Tsukuba, Ibaraki, Japan, 9 Laboratory of Plant Molecular Genetics, Faculty of Life Sciences, Kyoto Sangyo University, Kyoto, Japan, 10 The Graduate University for Advanced Studies, Research Center for Integrative Evolutionary Science, SOKENDAI, Hayama, Kanagawa, Japan

* terauchi@ibrc.or.jp

## Abstract

Dioecious plants, which have distinct male and female individuals, constitute ~5% of angiosperm species and have emerged frequently and independently from hermaphroditic ancestors. Although recent molecular studies of sex determination have started to reveal the diversity of the genetic systems underlying dioecy, research on the evolution of dioecy is limited, especially in monocots. Here, we explore the molecular basis of sex determination in the monocot *Dioscorea tokoro*, a dioecious wild yam endemic to East Asia. Chromosome-scale and haplotype-resolved genome assemblies and linkage analysis suggested that this plant has a male heterogametic sex-determination (XY) system, with sex-determination regions located on chromosome 3. Sequence comparison between the X- and Y-chromosomes and read coverage analysis revealed X- and Y-specific regions in putative pericentromeric chromosome regions. Within the Y-specific region, we propose two candidate genes that are likely involved in sex determination: *BLH9*, encoding a homeobox protein, and *HSP90*, encoding a molecular chaperone. BLH9 functions in a similar way as AtBLH9 in *Arabidopsis thaliana*. BLH9 could be involved in suppression of female organ development, whereas HSP90 might be required for pollen development. These results shed light on the complex evolution of dioecy in plants.

**Data availability statement:** All data are available in the Supplementary data and public databases (NCBi: https://www.ncbi.nlm.nih.gov/bioproject/PRJNA1223176) (zenodo: https://zenodo.org/records/14899024).

**Funding:** This study was supported by a JSPS KAKENHI Grant-in-Aid (https://www.jsps.go.jp/english/) for JSPS Fellows (23KJ1322 to AK), a Grant-in-Aid for Transformative Research Areas (23H04745 to RT), and JST PRESTO (https://www.jst.go.jp/kisoken/presto/en/) (JPMJPR22D2 to HK). The funders did not play any role in the study design, data collection and analysis, decision to publish, or preparation of the manuscript.

**Competing interests:** The authors have declared that no competing interests exist.

## Author summary

Sexual reproduction is a nearly universal, indispensable feature of evolution in eukaryotes. The molecular mechanisms underlying sex determination vary depending on the taxon. However, most information about this process was derived from studies of model organisms and/or domesticated species. To elucidate the diversity and evolution of sex determination, we need to expand the taxonomic breadth of these investigations. Here, we focus on the monocot genus *Dioscorea*, which contains species with multiple sex-determination systems, suggesting that frequent evolutionary transitions occur between them. We investigated the genetics of sex determination in *Dioscorea tokoro*, a dioecious wild yam endemic to East Asia. Whole-genome assembly and genetic analysis, along with transcriptome analysis, suggested that this species follows a male heterogametic sex-determination (XY) system, with two Y chromosome–specific genes that might be involved in male and female differentiation. These findings enhance our understanding of the complex evolution of dioecy.

## Introduction

Dioecy is a mating system in which male and female flowers are borne on separate individuals. This system enforces outcrossing and enables efficient pollen and seed production due to the specialization of male or female functions [1–3]. Approximately 15,600 dioecious plant species have been identified, representing 5–10% of angiosperm species. The sex phenotype of dioecious plant species is thought to be genetically controlled [4–6]. The wide and scattered taxonomic distribution of dioecious plants points to the frequent independent emergence of genetic systems controlling the male and female traits [7–9].

The sex phenotype is usually controlled by either male or female sex heterogametic system (XY/XX or ZZ/ZW). However, angiosperm species frequently lack heteromorphic sex chromosomes with detectable morphological differentiation. Recent advances in genome sequencing, however, made it possible to identify small sex-linked regions, which allowed the discovery of major sex-determining genetic components over the past decade [6]. In sex determination of angiosperm species, the Y or W chromosome is generally thought to carry one or two genetic components for male or female sex determination [3,10,11]. In Caucasian persimmon (*Diospyros lotus*) [12–14], *Oppressor of meGI* (*OGI*) on the Y chromosome encodes a small RNA that silences *Male Growth Inhibitor* (*MeGI*) and thereby promotes male development. Heterologous expression of *MeGI* in *Arabidopsis thaliana* and *Nicotiana tabacum* inhibits anther development, resulting in female flower development. Thus, in this plant, sex appears to be determined by a single genetic switch, *OGI* [12–15]. In garden asparagus (*Asparagus officinalis*), two genes on the Y chromosome, *SUPPRESSOR OF FEMALE FUNCTION* (*SOFF*) and *TAPETAL DEVELOPMENT AND FUNCTION1* (*aspTDF1*), function independently to suppress pistil development and promote

PLOS Genetics

anther development, respectively [16–19]. The genetic basis of sex determination has been revealed or predicted in ten other angiosperm genera. Variable genetic mechanisms for sex determination have been reported for different genera and even for different species within the same genus [6,20–22]. To understand the evolution of dioecy, the molecular basis of sex determination in a wide range of dioecious genera must be elucidated.

*Dioscorea* is the largest genus in the monocot family Dioscoreaceae and consists of approximately 630 species [23]. Most *Dioscorea* species are perennial herbaceous climbers that are widely distributed in tropical and temperate regions [24–26]. This genus includes the important tuber crop yam; reference genome sequences are available for four yam species [27–31]. Most *Dioscorea* species are dioecious, and multiple sex-determination systems have been proposed for different species in the genus based on cytological observations and genetic linkage analyses: a male heterogametic sex-determination system XY/XX in *D. tokoro*, *Dioscorea gracillima*, *Dioscorea bulbifera*, *Dioscorea dumetorum*, *Dioscorea tomentosa*, *Dioscorea pentaphylla*, *Dioscorea spinosa*, *Dioscorea alata*, and *Dioscorea japonica* [32–36]; a female heterogametic sex-determination system ZZ/ZW in *Dioscorea deltoidea* and *Dioscorea rotundata* [27,37]; extra chromosomes in females XO/XX in *Dioscorea sinuata* and *Dioscorea reticulata* [38,39]; and tetraploid sex chromosomes with a male heterogametic (XXYY or XXXY) and female homogametic (XXXX) system in *Dioscorea floribunda* [40]. Different sex-determination systems occur within individual sections (a taxonomic rank below the genus) of the genus. For example, the sections Stenophora and Enantiophyllum each contain species with XY/XX and ZZ/ZW systems [41,42], suggesting that genomic regions involved in sex determination arise via evolutionary transitions or independent evolution.

Recent genome analyses revealed genomic regions associated with sex phenotypes in *D. rotundata* and *D. alata*. Sex-linked genomic regions of *D. rotundata* were identified by quantitative trait locus sequencing (QTL-seq) analysis of $F_1$ progeny segregating for male and female plants [27]. A region of chromosome 11 in *D. rotundata* is associated with female heterogametic single-nucleotide polymorphism (SNP) markers and the presence of a female-specific (W-) genomic region was inferred, suggesting that sex determination in *D. rotundata* involves a ZZ/ZW system. Sex-linked regions of *D. alata* were identified by linkage analysis of $F_1$ progeny using SNP markers [35]. Linkage between DNA markers and sex was detected on the distal part of linkage group 6 in the male consensus genetic maps, suggesting that *D. alata* follows an XY/XX sex-determination system. A genome-wide association study in *D. alata* identified ~3-Mb sex-linked regions [43]. A recently assembled phased, telomere-to-telomere genome assembly of *D. alata* contains an~7.6-Mb potential sex-determination region (SDR) including a pericentric inversion [44]. The authors identified 88 sex-linked candidate genes and suggested that the jasmonic acid biosynthesis and signaling pathways may be involved in sex determination. However, the evolutionary relationship of the SDR across *Dioscorea* species is unclear. For instance, the *D. alata* SDR corresponds to chromosome 6 [35,44], while that of *D. rotundata* was identified on chromosome 11 [27], which does not share homologous genes and gene order similarity with *D. alata* chromosome 6. These findings strongly suggest that sex-determination systems have transitioned from one type to another during the evolution of the genus [35,42]. It is therefore important to clarify the evolutionary history of sex determination system in the genus. Which species maintain the ancestral state of the sex determination of the genus? How and when did the transitions happen from the ancestral state? Once we identify the SDRs, we may be able to identify the homologous genes located on the X- and Y-specific regions (XY gametologs) [45,46] and estimate their divergence by calculating their synonymous substitution rate to infer the evolutionary history of SDRs of the species. Also, in recent studies, stepwise sequence divergence patterns in the SDRs referred to as the evolutionary strata have been reported as the consequence of serial recombination suppression in the SDRs in animal and fungal sex chromosomes [47–50], as well as in plant sex chromosomes of *Silene latifolia* and its relative species [51–54] and hops [22]. Elucidating the trajectory of sex chromosome evolution and evolutionary strata of SDRs in the genus *Dioscorea* will also advance our understanding of the evolution of sex-determination systems.

In this study, we focused on the dioecious wild yam *D. tokoro* Makino (Figs 1A-1D and S1), a diploid species ($2n = 2x = 20$) that is widely distributed in temperate East Asia, including Japan, Korea, and China [23,55]. A previous DNA marker linkage study suggested that this species follows an XY/XX male heterogametic sex-determination system

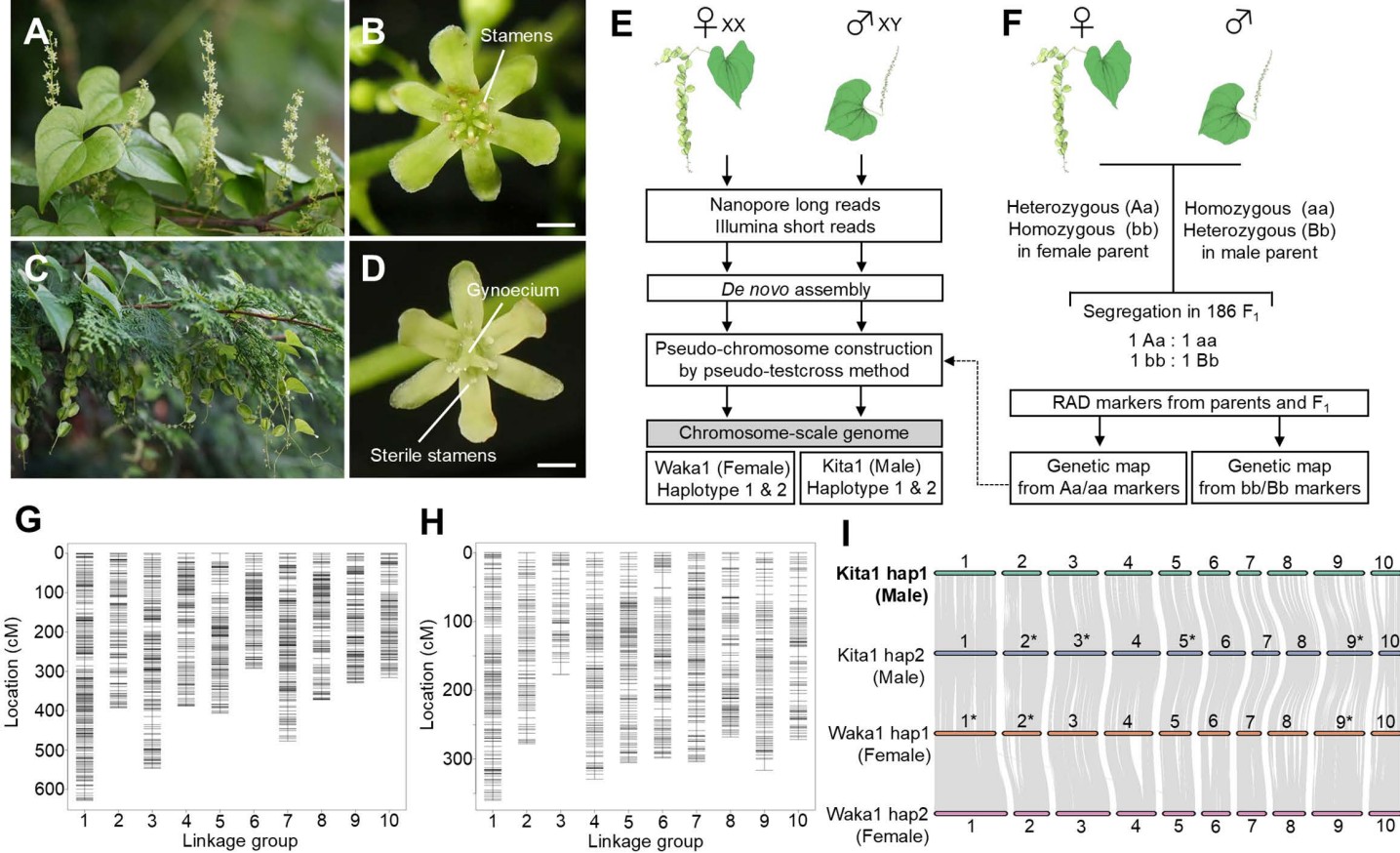

**Fig 1. Haplotype-resolved genome assemblies constructed from male and female individuals of the dioecious wild yam _D. tokoro._** Male and female flowers are borne on separate individuals in _D. tokoro_. **(A-D)** Male inflorescences **(A)**, a male flower with stamens **(B)**, female inflorescences with capsular fruits **(C)**, and a female flower with a gynoecium and sterile stamens **(D)**. Scale bars, 1 mm. **(E)** A simplified scheme for generating chromosome-scale female and male reference genomes using long reads generated by Oxford Nanopore Technologies and Illumina sequencing. **(F)** A simplified scheme for anchoring the assemblies using RAD-seq-based linkage maps generated by the pseudo-testcross method using 186 $F_1$ progenies. **(G, H)** Linkage maps based on female-parent-heterozygous markers **(G)** and male-parent-heterozygous markers **(H)**. **(I)** Chromosomal synteny of the two haplotypes of the female and male reference genomes. The Kita1 (male) haplotype 1 reference genome has the most robust foundation, with 10 chromosomes constructed from 18 contigs, and three other reference genomes showing high collinearity with this genome. The chromosomes marked with asterisks were subjected to reverse complementation. The illustrations of _D. tokoro_ were obtained with TogoTV (https://doi.org/10.7875/togopic.2022.552 and https://doi.org/10.7875/togopic.2022.453; copyright 2016 DBCLS TogoTV/ CC-BY-4.0).

[34]. Here, we obtained chromosome-scale and haplotype-resolved genome assemblies from female and male _D. tokoro_ plants. Association analysis of the $F_1$ progeny and a whole-genome comparison between female and male individuals revealed an SDR residing in the middle of chromosome 3 and containing X- and Y-specific regions. Based on transcriptome analysis, we identified the homeobox gene _BLH9_ and the molecular chaperone gene _HSP90_ as candidates for sex determination in _D. tokoro_.

## Results

### Reconstructing the genome sequence of _D. tokoro_ using genome assembly and linkage analysis

We reconstructed a chromosome-scale, haplotype-resolved genome assembly from female and male individuals of _D. tokoro_ using whole-genome sequencing reads obtained using the Oxford Nanopore Technologies (ONT) and Illumina

sequencing platforms (Fig 1E). The genome size of *D. tokoro* was estimated to be 388 Mb by flow cytometry analysis (S2 Fig). A previous linkage study suggested that this species follows an XY/XX male heterogametic sex-determination system [34]. Therefore, we generated separate assemblies from the filtered ONT reads for haplotypes 1 and 2 of a female individual (Waka1) collected from Tahara, Wakayama Pref., Japan, presumably with XX genotype. We also obtained separate assemblies using filtered ONT reads for haplotypes 1 and 2 of a male individual (Kita1) collected from Kitakami, Iwate Pref., Japan, presumably with XY genotype. For Waka1 individual (female), we obtained a haplotype 1 assembly with 212 contigs (425.1 Mb in total) and a haplotype 2 assembly with 346 contigs (342.2 Mb in total) after polishing the draft assemblies using Illumina reads (S1 Table). Likewise, for Kita1 individual (male), we obtained a haplotype 1 assembly with 172 contigs (414.4 Mb in total) and a haplotype 2 assembly with 440 contigs (300.4 Mb in total). Notably, the Waka1 haplotype 1 contained two telomere-to-telomere contig assemblies, and the Kita1 haplotype 1 contained six telomere-to-telomere contig assemblies. BUSCO analysis [56] showed that the percentages of complete embryophyte BUSCOs were 98.2% in the Waka1 haplotype 1 assembly, 81.9% in the Waka1 haplotype 2 assembly, 98.0% in the Kita1 haplotype 1assembly, and 76.0% in the Kita1 haplotype 2 assembly (S1 Table). The relatively low sizes of the haplotype 2 assemblies of Waka1 and Kita1 were probably caused by using the PECAT program, which prioritizes the assembly of haplotype 1; similar biases were previously reported [57].

To anchor the assembled sequences to chromosomes by linkage analysis, we used a pseudo-testcross approach [58] (Fig 1F), a method to address the linkage of two DNA markers, both of which are heterozygous in one parent and homozygous in another parent, by testing their co-segregation or independent segregation in the $F_1$ progeny population (S3 Fig). We obtained 186 $F_1$ progeny from a cross between Waka1 (female) and Kita1 (male) (S4 Fig) and genotyped these progeny and the two parent plants by restriction site–associated DNA sequencing (RAD-seq) [59]. We selected SNP and presence/absence polymorphism (PA) markers that were heterozygous in the female parent and homozygous in the male parent as female-parent-heterozygous markers (S5 Fig). We also obtained SNP and PA markers that were homozygous in the female parent and heterozygous in the male parent as male-parent-heterozygous markers. Using the pseudo-testcross scheme, we constructed two separate linkage maps using the female-parent-heterozygous markers and the male-parent-heterozygous markers (Figs 1G, 1H; S6, S7, S8, and S9). Finally, we combined the two linkage groups using the shared assemblies between the two linkage maps and generated pseudo-chromosomes 1–10 (S10, S11, S12, and S13 Figs); for simplicity, we refer to the pseudo-chromosomes as chromosomes hereafter. The chromosome-assigned sequences using the Waka1 (female) haplotype 1 and Kita1 (male) haplotype 1 references appears to be close to complete, with 10 chromosomes containing only 24 and 18 contigs, respectively (S1 Table).

To annotate the gene models, we obtained RNA-seq data from 18 different organs of *D. tokoro* and used protein homology information for *D. alata* TDa95/00328 and *D. rotundata* TDr96_F1. Based on the combination of transcriptome-based gene identification and *ab initio* gene prediction, we identified 21,308 genes for the Waka1 (female) haplotype 1 assembly, 17,710 genes for the Waka1 (female) haplotype 2 assembly, 23,200 genes for the Kita1 (male) haplotype 1 assembly, and 18,145 genes for the Kita1 (male) haplotype 2 assembly (S2 Table). Chromosomal synteny analysis based on collinear blocks with predicted genes revealed high collinearity between the Kita1 (male) haplotype 1–based chromosomes with the largest number of telomere-to-telomere assemblies and the chromosomes inferred using the three other reference genomes (Fig 1I). We determined that the chromosome-scale genome assemblies obtained from female and male individuals are useful to infer the structures of X- and Y-specific genomic regions.

### Linkage study and mapping coverage analysis of Illumina short reads identify the X- and Y-specific genomic regions on chromosome 3

To identify the genomic region linked to the sex phenotype, we performed association analysis using the 186 $F_1$ progeny, comprising 38 females, 89 males, and 59 non-flowering individuals. Based on RAD-seq data from the 127 flowering $F_1$ progeny, we examined the association between the genotypes and sex phenotype of the $F_1$ individuals using Fisher's

exact tests (Figs 2A and S14). The log-transformed $q$-values ($-\log_{10}(q)$) revealed significant associations between sex phenotype and the male-parent-heterozygous markers in the middle of chromosome 3 but did not detect any association using the female-parent-heterozygous markers (Figs 2B, 2C, and S15). These associations were also confirmed by a Fisher's exact test using the same number of females and males randomly selected from the $F_1$ progeny (S16 Fig). Linkage analysis using simple interval mapping performed with R/qtl [60] also detected linkage between sex phenotype and the male-parent-heterozygous markers on chromosome 3, whereas no linkage was detected with the female-parent-heterozygous markers (S17 Fig). These results suggest that the sex of *D. tokoro* is determined by components on chromosome 3 and that the species has a male heterogametic sex-determination system of XY/XX.

In the genomic region associated with sex phenotype, we detected a higher concentration of presence/absence polymorphism (PA) markers (358 markers in the 10-Mbp region around the marker showing the peak association value) than SNP markers (62 markers in the 10-Mbp region) (Fig 2B and 2C), indicating structural differences in the middle of chromosome 3. Alignment-based dot plot analysis of chromosome 3 between the female and male haplotype 1 assemblies detected a ~5 Mb mismatch in the middle of the chromosome (Fig 2D), confirming the differentiation of chromosome 3. The chromosome 3 assemblies corresponding to the female haplotype 1 and the male haplotype 1 contained only three and two contigs, respectively, with the mismatch region located in the middle of contigs in both assemblies (Figs 2D, C, and D in S1 Text), indicating that the observed structural differentiation is reliable. Therefore, we further examined the structural differences in this region by performing coverage analysis of Illumina short-read sequences of the male and female genomes mapped to the reference genome sequences. In diploid plants with male heterogametic sex determination (XY) and substantial structural differentiation between X and Y chromosomes, the Y-specific region is predicted to be absent from females. Furthermore, the coverage of the male X-specific region is thought to be one-half that of females (hemizygous) [61,62]. To assess whether this is the case for *D. tokoro*, we aligned Illumina short reads obtained from female and male individuals to each haplotype of chromosome 3 and studied the mapping depth. As the reference genome sequences for mapping, we used four different chromosome 3 assemblies: the Waka1 (female) haplotype 1 and 2 assemblies and the Kita1 (male) haplotype 1 and 2 assemblies. We identified a region in the middle of chromosome 3 that was enriched for sites covered by male reads at 0.5×depth and female reads at 1×depth in the Waka1 (female) haplotype 1 assembly (Fig 2E). Such consistent depth differences between male and female reads were not observed in other chromosomes (S18 Fig). When we mapped the Illumina short reads to the Kita1 (male) haplotype 1 assembly, we identified a region in the middle of chromosome 3 that was enriched for sites covered by male reads at 0.5×depth and female reads at 0×depth (Fig 2F). Such depth differences between male and female reads were not observed in other chromosomes (S19 Fig). We also observed such differences in depth between male and female reads when male and female plants collected from three different locations were used for mapping (S20 and S21 Figs). In contrast, we did not detect regions covered by male reads at 0.5×depth and female reads at 1×depth in the center of chromosome 3 of the Waka1 (female) haplotype 2 (S22A Fig), possibly due to incomplete assembly of haplotype 2. We detected a region covered by male reads at 0.5×depth and female reads at 1×depth in the Kita1 (male) haplotype 2 assembly (S22B Fig).

To confirm the putative SDRs identified by the association study and mapping coverage analysis, we applied two previously reported methods: SDpop, which detects SDRs by modelling the allele and genotyping frequencies for different segregation types [63], and RADSex, which detects SDRs based on the association of RAD markers with male and female individuals [64]. For SDpop analysis, we used Illumina short read sequences from five females and five males of *D. tokoro* collected from a natural population in Wakayama prefecture, central Japan. Based on the alignments of these sequences to each haplotype reference, empirical Bayes posterior probabilities in the SDpop model inferred three segregation types at each polymorphic site: autosomal segregation type, X-hemizygosity segregation type and XY gametology segregation type. High posterior probabilities of XY gametology segregation type and low posterior probabilities of autosomal segregation type were detected in the middle of chromosome 3 of both the Waka1 (female) haplotype 1 assembly and the Kita1 (male) haplotype 1 assembly. These patterns suggest the presence of SDRs at positions consistent with

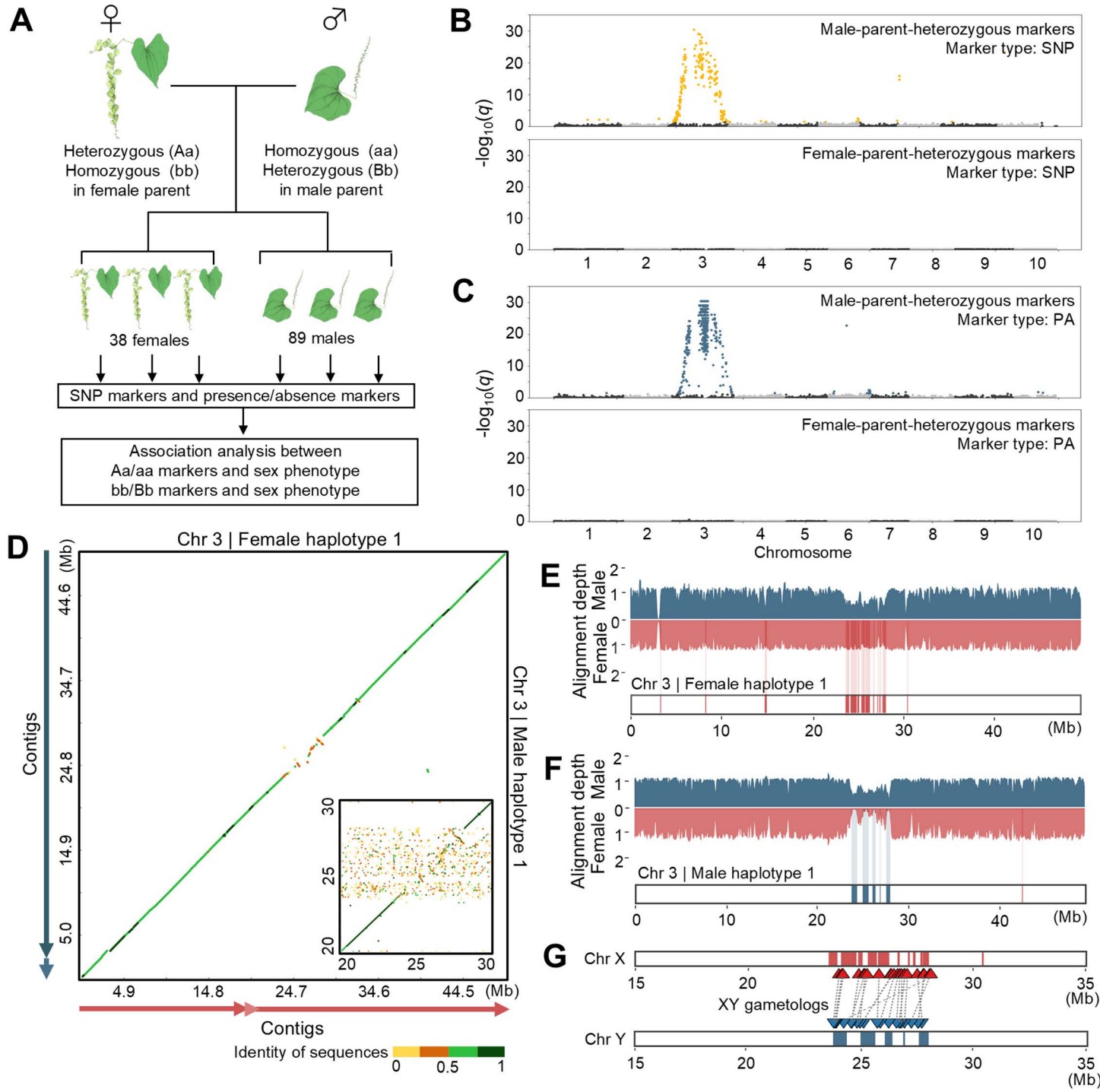

**Fig 2. The dioecious wild yam *D. tokoro* contains X- and Y-specific regions on chromosome 3. (A)** Schematic diagram of association analysis to identify genomic regions associated with sex phenotype. Single-nucleotide polymorphism (SNP) and presence/absence (PA) markers were detected in 38 females and 89 males in the $F_1$ progeny. **(B)** Manhattan plots of $-\log_{10}(q)$ values between SNP-type markers and sex phenotype based on the Kita1 (male) haplotype 1 assembly. The $-\log_{10}(q)$ values were obtained by Fisher's exact test. The upper and lower plots show male-parent-heterozygous and female-parent-heterozygous markers, respectively. The yellow points in the upper plot indicate SNP-type markers significantly associated with sex phenotype. A significance threshold of 5% false discovery rate was adjusted by Benjamini-Hochberg correction: adjusted threshold = 1.31 for plots of male-parent-heterozygous markers and adjusted threshold = 2.00 for plots of female-parent-heterozygous markers. **(C)** Manhattan plots

of $-\log_{10}(q)$ values between PA markers and sex phenotype based on the Kita1 (male) haplotype 1 assembly. The red and blue plots show PA markers significantly associated with sex phenotype. The male-parent-heterozygous markers in the middle of chromosome 3 show significant associations. **(D)** Alignment-based dot plot analysis of chromosome 3 between the Waka1 (female) haplotype 1 and the Kita1 (male) haplotype 1 showed low sequence matches in the middle of the chromosome. Blue arrows in the left and red arrows in the bottom indicate positions of the contigs for Kita1 (male) haplotype 1 and Waka1 (female) haplotype 1, respectively. Waka1 (female) haplotype 1 and Kita1 (male) haplotype 1 assemblies corresponding to chromosome 3 contained only three and two contigs, respectively, with the mismatch region located in the middle of contigs in both assemblies (Figs C and D in S1 Text). The inset panel shows a zoomed-in alignment of 20-30 Mb region of the chromosomes, including the mismatch region. The alignment-based dot plots were generated using D-Genies, a software performing large-scale genome alignment visualized as dot plots [131]. Chromosome 3 of Kita1 (male) haplotype 1 was used as the target sequence, and chromosome 3 of Waka1 (female) haplotype 1 was used as the query sequence. The full plot was generated with noise filtering, and the inset panel was generated without noise filtering, using the D-Genies algorithm that removes small matches based on the alignment size and frequency. **(E)** Mapping coverage analysis to differentiate between putative X- and Y-specific regions. The bar graphs indicate female and male mapping depth, obtained by mapping Waka1 (female) and Kita1 (male) Illumina sequences onto the haplotype 1 assembly from Waka1 (female). Coverage analysis revealed putative X-specific regions of chromosome 3, shown as red regions below the bar graphs. Waka1 haplotype 1 chromosome 3 was defined as the X chromosome. **(F)** Bar graphs showing female and male mapping depth, obtained by aligning Waka1 (female) and Kita1 (male) Illumina sequences onto the haplotype 1 assembly from Kita1 (male). Coverage analysis revealed putative Y-specific regions on chromosome 3, shown as blue regions below the bar graphs. Kita1 haplotype 1 chromosome 3 was defined as the Y chromosome. **(G)** Positions of XY gametologs on the putative X- and Y- specific regions. The illustrations of *D. tokoro* were obtained with TogoTV (https://doi.org/10.7875/togopic.2022.552 and https://doi.org/10.7875/togopic.2022.453; copyright 2016 DBCLS TogoTV/ CC-BY-4.0).

the X- specific and Y-specific regions identified by the mapping coverage analysis (S23A and S23B Fig). For RADSex analysis, we used RAD markers of 127 $F_1$ flowering progeny from a cross between Waka1 (female) and Kita1 (male). The probability of association with sex, $-\log_{10}(p)$, was calculated for each marker by a Pearson's chi-squared test between marker and sex phenotype in RADSex. Sex-associated markers with high $-\log_{10}(p)$ values were detected in chromosome 3 of both the Kita1 (male) haplotype 1 assembly (S24A Fig) and the Waka1 (female) haplotype 1 assembly (S24B Fig). These regions corresponded to the SDRs identified by the mapping coverage analysis (S24C and S24D Fig).

Based on the results of our association analysis (Fig 2B and 2C), alignment-based dot plot analysis (Fig 2D), mapping coverage analysis (Fig 2E and 2F) and the SDR detection analyses (S23 and S24 Figs), we define chromosome 3 of the Waka1 (female) haplotype 1 assembly as the X chromosome and chromosome 3 of the Kita1 (male) haplotype 1 assembly as the Y chromosome. These chromosomes are differentiated into putative X- and Y-specific regions spanning ~4.8 Mb with 54 candidate genes (S3 Table) and ~4.5 Mb with 56 candidate genes (Table 1), respectively. These candidate genes include X- or Y-specific genes and putative orthologous genes (gametologs; [45,46]). A total of 22 genes in the Y- region had corresponding genes in the X-region so that they were defined as XY gametologs (Fig 2G).

## X- and Y-specific regions are likely located in the pericentromeric regions

Recombination suppression is a key step in SDR evolution. This suppression is caused by various mechanisms, such as chromosome rearrangements including inversions and transpositions [6,65,66]. Recombination suppression has also been reported even in the absence of structural rearrangements, accompanied by gene loss and accumulation of repetitive DNA sequences [67,68]. Accumulation of repetitive DNA in the non-recombining regions (NRY) surrounding the SDR are reported as a conspicuous feature of sex chromosomes [69–73]. In *D. tokoro*, the alignment-based dot plot analysis showed no major inversion between the X and Y chromosomes (Fig 2D). The X- and Y-specific regions revealed to be located in a genomic region with lower gene density and higher density of repetitive DNA sequences, represented by Ty3 long terminal repeats (LTRs) (Fig 3A and 3B). Co-localization of X- and Y-specific regions to the gene-paucity and repetitive DNA-rich regions suggests that *D. tokoro* X- and Y-specific regions likely correspond to the pericentromeric regions (near the centromere) of chromosome 3.

We also evaluated the recombination rate around the X- and Y-specific regions by comparing the relationship between the linkage distances (in centimorgans, cM) and physical distances (in base pairs, bp) (S10 and S12 Figs). Linkage distance, recombination frequency between a pair of DNA markers on a chromosome, was obtained based on the

Table 1. Differential expression analysis of genes located in Y-specific regions.

| No. | Name | Y specificity | Putative homologues in *D. tokoro* genome | base-Mean | Adjusted p-value | log2FC | base-Mean | Adjusted p-value | log2FC | Top hit gene | Gene accession | E-value | Pident | Sequence overlap (bp) |
|---|---|---|---|---|---|---|---|---|---|---|---|---|---|---|
| | | Y specificity of the genes | | Male (S0, S1, S2) vs. female (S0, S1, S2) | | | Male (S0, S1, S2) vs. non-reproductive organs | | | BLASTX (Swiss-Prot) | | | | |
| 1 | DtKi-ta1h1_04777.t1.p1 (*BLH9*) | Y-specific | | 317.5 | 1.20E-63 | 5.44 | 143.8 | 9.10E-31 | 5.69 | BEL1-like homeodomain protein 9 \| *Arabidopsis thaliana* | Q9LZM8.1 | 6.50E-23 | 44.5 | 137 |
| 2 | DtKi-ta1h1_04754.t1.p1 | Y-specific | | 501.3 | 1.90E-21 | -2.18 | 484.7 | 0.0017 | -2.06 | Putative nuclease HARBI1 \| *Bos taurus* | Q17QR8.1 | 1.60E-10 | 27.6 | 203 |
| 3 | DtKi-ta1h1_04791.t1.p1 | Y-specific | | 627.2 | 5.88E-11 | 1.11 | 624.3 | 0.42 | 0.21 | Zinc finger MYM-type protein 1 \| *Homo sapiens* | Q5SVZ6.1 | 4.80E-33 | 25.2 | 317 |
| 4 | DtKi-ta1h1_04764.t1 | Y-specific | | 62.9 | 6.60E-11 | 7.53 | 145.7 | 0.70 | -0.68 | | | | | |
| 5 | DtKi-ta1h1_04801.t1.p1 (*HSP90*) | Y-specific | | 39.6 | 8.20E-11 | 8.73 | 24.8 | 0.0014 | 2.70 | Heat shock protein 81–1 \| *Oryza sativa* Indica Group | A2YWQ1.1 | 3.94E-14 | 60.0 | 50 |
| 6 | DtKi-ta1h1_04799.t1.p1 | Y-specific | | 62.2 | 4.40E-10 | 7.51 | 75.7 | 0.68 | 0.59 | | | | | |
| 7 | DtKi-ta1h1_04768.t1 | Y-specific | | 27.5 | 2.30E-09 | 8.20 | 40.9 | 0.88 | 0.17 | | | | | |
| 8 | DtKi-ta1h1_04787.t1.p1 | Y-specific | | 21.7 | 2.40E-07 | 4.36 | 25.4 | 0.61 | 0.59 | | | | | |
| 9 | DtKi-ta1h1_04782.t1.p1 | Y-specific | | 12.5 | 1.80E-06 | 7.06 | 9.0 | 0.11 | 2.08 | | | | | |
| 10 | DtKi-ta1h1_04746.t1.p1 | Y-specific | | 12.1 | 5.10E-06 | 7.02 | 69.1 | 0.14 | -2.17 | | | | | |
| 11 | DtKi-ta1h1_04794.t1.p1 | Y-specific | | 141.1 | 1.60E-05 | -1.09 | 98.4 | 0.31 | -0.56 | Mitochondrial phosphate carrier protein 3, mitochondrial \| *Arabidopsis thaliana* | Q9FMU6.1 | 2.40E-42 | 91.5 | 59 |
| 12 | DtKi-ta1h1_04797.t1.p1 | Y-specific | | 6.7 | 2.80E-05 | 6.16 | 6.1 | 0.26 | 1.37 | | | | | |
| 13 | DtKi-ta1h1_04792.t1.p1 | Y-specific | | 8.7 | 3.90E-05 | 6.54 | 8.7 | 0.36 | 1.09 | | | | | |

*(Continued)*

Table 1. (Continued)

| Genes | | Y specificity of the genes | | Male (S0, S1, S2) vs. female (S0, S1, S2) | | | Male (S0, S1, S2) vs. non-reproductive organs | | | BLASTX (Swiss-Prot) | | | | |
|---|---|---|---|---|---|---|---|---|---|---|---|---|---|---|
| No. | Name | Y specificity | Putative homologues in *D. tokoro* genome | base-Mean | Adjusted *p*-value | log$_2$FC | base-Mean | Adjusted *p*-value | log$_2$FC | Top hit gene | Gene accession | *E*-value | Pident | Sequence overlap (bp) |
| 14 | DtKita1h1_04755.t1.p1 | Y-specific | | 52.5 | 0.047 | -0.85 | 22.9 | 0.56 | 0.60 | | | | | |
| 15 | DtKita1h1_04793.t1.p3 | Y-specific | | 44.1 | 0.078 | -0.80 | 19.9 | 0.50 | 0.56 | | | | | |
| 16 | DtKita1h1_04786.t1.p1 | Y-specific | | 487.7 | 0.083 | -0.32 | 482.5 | 0.30 | -0.60 | Sodium/calcium exchanger NCL1 | *Oryza sativa* Japonica Group | Q5QNI2.1 | 8.40E-46 | 73.3 | 105 |
| 17 | DtKita1h1_04769.t1.p4 | Y-specific | | 6.3 | 0.12 | -2.16 | 1.7 | 0.96 | 0.20 | | | | | |
| 18 | DtKita1h1_04770.t1.p1 | Y-specific | | 8.3 | 0.13 | -1.87 | 1.5 | 0.81 | 1.57 | | | | | |
| 19 | DtKita1h1_04758.t1.p1 | Y-specific | | 94.6 | 0.22 | 0.47 | 43.6 | 0.04 | 1.73 | Adenine phosphoribosyltransferase 1 | *Triticum aestivum* | Q43199.1 | 6.20E-22 | 44.2 | 129 |
| 20 | DtKita1h1_04776.t1.p1 | Y-specific | | 72.7 | 0.28 | -0.38 | 31.0 | 0.064 | 1.12 | | | | | |
| 21 | DtKita1h1_04767.t1.p1 | Y-specific | | 3.6 | 0.51 | 1.05 | 6.2 | 0.69 | -0.83 | Protein RICE SALT SENSITIVE 3 | *Oryza sativa* Japonica Group | K4PW38.1 | 2.40E-12 | 52.3 | 65 |
| 22 | DtKita1h1_04785.t1 | Y-specific | | 195.1 | 0.73 | -0.10 | 215.9 | 0.14 | -0.65 | Sodium/calcium exchanger NCL2 | *Oryza sativa* Japonica Group | Q6K3R5.2 | 2.20E-14 | 82.6 | 46 |
| 23 | DtKita1h1_04779.t1.p1 | Y-specific | | 7.8 | 0.99 | 0.01 | 7.1 | 0.85 | -0.22 | Probable leucine-rich repeat receptor-like protein kinase At2g33170 | *Arabidopsis thaliana* | O49318.1 | 8.80E-15 | 49.2 | 59 |
| 24 | DtKita1h1_04795.t1 | Y-specific | | 1.5 | NA | 2.90 | 1.5 | 0.72 | 0.92 | | | | | |

*(Continued)*

Table 1. (Continued)

| Genes | | Y specificity of the genes | | Male (S0, S1, S2) vs. female (S0, S1, S2) | | | Male (S0, S1, S2) vs. non-reproductive organs | | | BLASTX (Swiss-Prot) | | | | |
|---|---|---|---|---|---|---|---|---|---|---|---|---|---|---|
| No. | Name | Y specificity | Putative homologues in *D. tokoro* genome | base-Mean | Adjusted p-value | log$_2$FC | base-Mean | Adjusted p-value | log$_2$FC | Top hit gene | Gene accession | E-value | Pident | Sequence overlap (bp) |
| 25 | DtKita1h1_04762.t1 | Y-specific | | 0.7 | NA | 1.40 | 0.2 | 0.77 | 1.95 | Transposon Tf2–1 polyprotein \| *Schizosaccharomyces pombe* 972h- | P0CT34.1 | 5.30E-07 | 29.6 | 135 |
| 26 | DtKita1h1_04788.t1.p1 | Y-specific | | 0.2 | NA | 0.92 | 0.1 | 0.95 | 0.52 | | | | | |
| 27 | DtKita1h1_04773.t1 | Y-specific | | 0.2 | NA | 0.92 | 0.2 | 0.98 | 0.15 | | | | | |
| 28 | DtKita1h1_04751.t1 | Y-specific | | 0.5 | NA | 0.88 | 0.3 | 0.92 | 0.73 | | | | | |
| 29 | DtKita1h1_04790.t1 | Y-specific | | 1.7 | NA | 0.60 | 3 | 0.75 | -1.06 | Glutamate receptor 1.4 \| *Arabidopsis thaliana* | Q8LGN1.2 | 1.10E-18 | 33.3 | 132 |
| 30 | DtKita1h1_04752.t1 | Y-specific | | 0 | NA | NA | 0 | NA | NA | | | | | |
| 31 | DtKita1h1_04753.t1 | Y-specific | | 0 | NA | NA | 0 | NA | NA | | | | | |
| 32 | DtKita1h1_04756.t1 | Y-specific | | 0 | NA | NA | 0 | NA | NA | | | | | |
| 33 | DtKita1h1_04765.t1 | Y-specific | | 0 | NA | NA | 0.2 | 0.87 | -1.17 | | | | | |
| 34 | DtKita1h1_04771.t1 | Y-specific | | 0 | NA | NA | 0.1 | 0.91 | -0.81 | | | | | |
| 35 | DtKita1h1_04772.t1 | Y-specific | | 0 | NA | NA | 0 | NA | NA | | | | | |
| 36 | DtKita1h1_04759.t1.p1 | XY gametolog | DtWaka1_05806.t1.p1 | 1705.0 | 2.05E-25 | -1.26 | 1391.8 | 0.16 | -0.98 | | | | | |
| 37 | DtKita1h1_04749.t1.p1 | XY gametolog | DtWaka1_05796.t2.p1 | 28.2 | 1.35E-09 | 8.24 | 24.1 | 0.18 | 1.49 | | | | | |

*(Continued)*

Table 1. (Continued)

| Genes | | Y specificity of the genes | | Male (S0, S1, S2) vs. female (S0, S1, S2) | | | Male (S0, S1, S2) vs. non-reproductive organs | | | BLASTX (Swiss-Prot) | | | | |
|---|---|---|---|---|---|---|---|---|---|---|---|---|---|---|
| No. | Name | Y specificity | Putative homologues in *D. tokoro* genome | baseMean | Adjusted *p*-value | log$_2$FC | baseMean | Adjusted *p*-value | log$_2$FC | Top hit gene | Gene accession | E-value | Pident | Sequence overlap (bp) |
| 38 | DtKita1h1_04766.t1.p1 | XY gametolog | DtWaka1_05808.t1.p1 | 847.8 | 1.84E-05 | -0.50 | 538.2 | 0.90 | 0.11 | Ribonucleoside-diphosphate reductase small chain A | *Arabidopsis thaliana* | P50651.2 | 1.60E-22 | 93.0 | 43 |
| 39 | DtKita1h1_04748.t1.p1 | XY gametolog | DtWaka1_05797.t1.p1 | 35.2 | 0.00040 | 1.97 | 64.7 | 0.75 | -0.66 | Heat stress transcription factor B-1 | *Oryza sativa* Japonica Group | Q67TP9.1 | 9.70E-28 | 85.5 | 62 |
| 40 | DtKita1h1_04796.t1.p1 | XY gametolog | DtWaka1_05831.t1.p1 | 876.0 | 0.0012 | 0.34 | 737.1 | 0.68 | 0.14 | | | | | |
| 41 | DtKita1h1_04778.t1.p1 | XY gametolog | DtWaka1_05810.t1.p1 | 675.4 | 0.0029 | -0.59 | 489.4 | 0.60 | -0.22 | Preprotein translocase subunit SCY2 | *Arabidopsis thaliana* | F4IQV7.1 | 6.10E-29 | 88.9 | 45 |
| 42 | DtKita1h1_04774.t1.p1 | XY gametolog | DtWaka1_05809.t1.p1 | 339.2 | 0.012 | -0.40 | 245.1 | 0.86 | -0.07 | Retrovirus-related Pol polyprotein from transposon RE1 | *Arabidopsis thaliana* | Q94HW2.1 | 3.30E-77 | 45.9 | 181 |
| 43 | DtKita1h1_04775.t1.p1 | XY gametolog | DtWaka1_05816.t1.p1 | 24.4 | 0.013 | -2.48 | 100.4 | 0.027 | -4.50 | Exocyst complex component EXO70C1 | *Arabidopsis thaliana* | Q9FY95.1 | 2.10E-175 | 47.0 | 593 |
| 44 | DtKita1h1_04781.t1.p1 | XY gametolog | DtWaka1_05821.t1.p1 | 481.9 | 0.093 | -0.28 | 349.1 | 0.95 | 0.020 | Transposon Ty3-I Gag-Pol polyprotein | *Saccharomyces cerevisiae* S288C | Q7LHG5.2 | 2.00E-30 | 35.7 | 182 |
| 45 | DtKita1h1_04763.t1.p1 | XY gametolog | DtWaka1_05807.t1.p1 | 526.9 | 0.12 | -0.26 | 517.8 | 0.25 | -0.55 | Peroxisomal adenine nucleotide carrier 1 | *Glycine max* | B6ZJZ9.1 | 2.30E-25 | 60.4 | 91 |
| 46 | DtKita1h1_04800.t1.p1 | XY gametolog | DtWaka1_05835.t1.p1 | 558.0 | 0.15 | -0.35 | 774.4 | 0.056 | -1.21 | Phosphatidylinositol 4-phosphate 5-kinase 1 | *Oryza sativa* Japonica Group | Q6EX42.2 | 1.86E-88 | 51.5 | 369 |
| 47 | DtKita1h1_04798.t1.p1 | XY gametolog | DtWaka1_05834.t1.p1 | 840.0 | 0.24 | 0.16 | 828.7 | 0.57 | -0.28 | | | | | |

*(Continued)*

Table 1. (Continued)

| Genes | | Y specificity of the genes | | Male (S0, S1, S2) vs. female (S0, S1, S2) | | | Male (S0, S1, S2) vs. non-reproductive organs | | | BLASTX (Swiss-Prot) | | | | |
|---|---|---|---|---|---|---|---|---|---|---|---|---|---|---|
| No. | Name | Y specificity | Putative homologues in D. tokoro genome | baseMean | Adjusted p-value | log$_2$FC | baseMean | Adjusted p-value | log$_2$FC | Top hit gene | Gene accession | E-value | Pident | Sequence overlap (bp) |
| 48 | DtKita1h1_04750.t1.p1 | XY gametolog | DtWaka1_05792.t1.p1 | 689.4 | 0.24 | -0.18 | 407.1 | 0.44 | 0.52 | Phosphatidylinositol 4-phosphate 5-kinase 1 | Arabidopsis thaliana | Q56YP2.1 | 1.10E-163 | 60.7 | 410 |
| 49 | DtKita1h1_04761.t1.p1 | XY gametolog | DtWaka1_05803.t1.p1 | 624.1 | 0.45 | 0.14 | 495.9 | 0.65 | 0.13 | 3',5'-bisphosphate nucleotidase AHL | Arabidopsis thaliana | Q38945.1 | 1.90E-84 | 63.0 | 230 |
| 50 | DtKita1h1_04783.t1.p1 | XY gametolog | DtWaka1_05826.t1.p1 | 185.0 | 0.48 | 0.17 | 159.5 | 0.96 | -0.02 | | | | | |
| 51 | DtKita1h1_04760.t1.p1 | XY gametolog | DtWaka1_05804.t1.p1 | 2.9 | 0.57 | 1.24 | 15.6 | 0.19 | -2.64 | | | | | |
| 52 | DtKita1h1_04747.t1.p2 | XY gametolog | DtWaka1_05800.t1.p2 | 960.0 | 0.73 | -0.13 | 705.3 | 0.97 | 0.10 | Aspartyl protease AED3 | Arabidopsis thaliana | O04496.1 | 1.90E-133 | 59.4 | 394 |
| 53 | DtKita1h1_04784.t1.p1 | XY gametolog | DtWaka1_05825.t1.p1 | 546.6 | 0.87 | -0.04 | 912.0 | 0.0067 | -1.32 | Protein WEAK CHLOROPLAST MOVEMENT UNDER BLUE LIGHT 1 | Arabidopsis thaliana | O48724.1 | 1.20E-168 | 57.8 | 614 |
| 54 | DtKita1h1_04757.t1.p1 | XY gametolog | DtWaka1_05802.t1.p1 | 24.8 | 0.99 | -0.01 | 314.5 | 9.60E-05 | -4.41 | Beta-galactosidase 6 | Oryza sativa Japonica Group | Q10NX8.2 | 4.60E-43 | 40.3 | 243 |
| 55 | DtKita1h1_04780.t1.p1 | XY gametolog | DtWaka1_05823.t1.p1 | 0.2 | NA | 0.92 | 0.8 | 0.78 | -1.91 | Aluminum-activated malate transporter 2 | Arabidopsis thaliana | Q9SJE8.2 | 6.40E-28 | 46.9 | 98 |
| 56 | DtKita1h1_04789.t1 | XY gametolog | DtWaka1_05791.t1.p1 | 0.5 | NA | -0.93 | 0.3 | 0.94 | -0.56 | Retrovirus-related Pol polyprotein from transposon RE2 | Arabidopsis thaliana | Q9ZT94.1 | 7.10E-127 | 41.6 | 476 |

The gray highlights indicate p-values <0.05. Blue highlights indicate log$_2$FC>2, representing genes highly expressed in male flowers in the comparison of male (S0, S1, S2) vs. female (S0, S1, S2) and the comparison of male (S0, S1, S2) vs. non-reproductive organs. Y specificity was defined based on putative orthologous or paralogous gene pairs identified from comparisons between female and male individuals of D. tokoro. Top hits of putative homologues in D. tokoro genome were obtained by BLASTN with the threshold of E-value <1 × 10$^{-100}$.

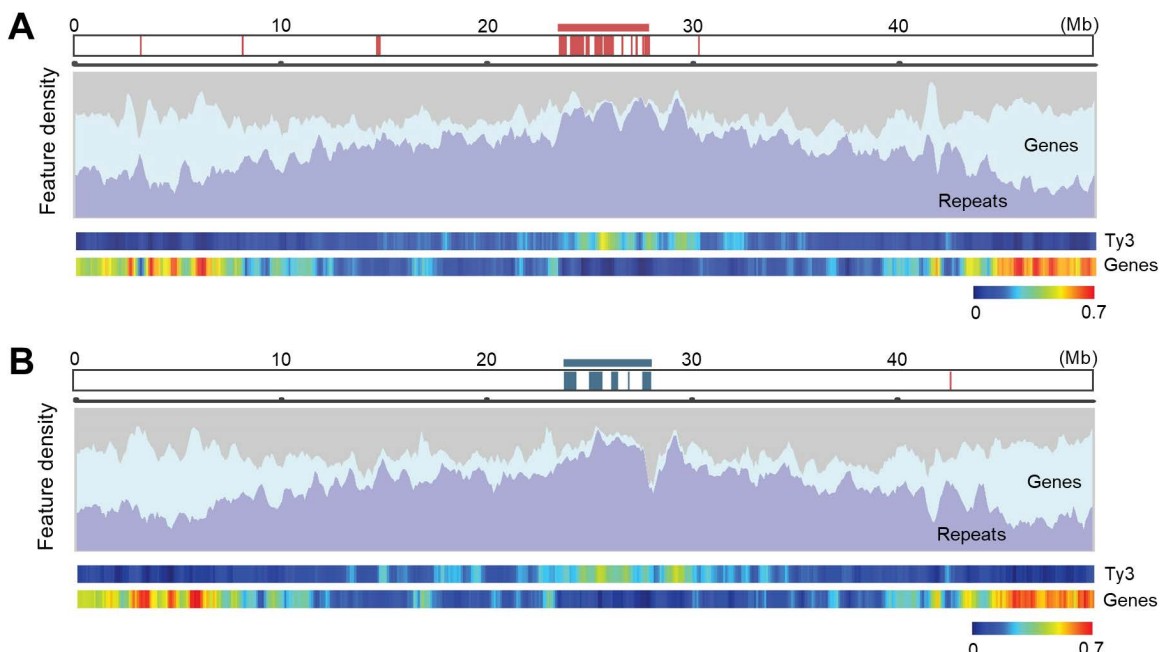

**Fig 3. X- and Y-specific regions locate in putative pericentromeric regions. (A)** Repetitive DNA sequences accumulate in X-specific regions on chromosome X (chromosome 3 of Waka1 haplotype 1 assembly), suggesting that these regions are located in the pericentromeric regions. The bar graphs indicate gene and repeat density on chromosome 3. The heatmap indicates gene and Ty3 retrotransposon density. Distributions of genes and repeat densities along the chromosomes were calculated by a sliding window analysis (window size = 500 kb and step size = 100 kb). For each window, the percentage bp coverage of the gene or repetitive element sequence was calculated with the number of non-N basepairs as denominators. Ty3 element belongs to "Repeats" category. **(B)** Repetitive DNA sequences accumulate in Y-specific regions of chromosome Y (chromosome 3 of Kita1 haplotype 1 assembly), suggesting that these regions are located in pericentromeric regions.

segregation of RAD markers among the $F_1$ progeny. Although we observed a small reduction in linkage distances as compared to physical distances in the center of chromosome 3, the pattern of relationship between linkage and physical distances was not greatly different from those of other chromosomes (S10 and S12 Figs). This suggests that the NRYs, even if exist, may be restricted to the small X- and Y-specific regions near the centromere with suppressed recombination, while most part of chromosome 3 constitutes pseudoautosomal regions (PARs) undergoing recombination. Based on these observations, we suggest that centromere-mediated recombination suppression has contributed to the evolution of *D. tokoro* X- and Y-specific regions possibly located in the vicinity of the centromere.

Most *Dioscorea* species are dioecious. However, it remains unclear whether the identified sex determination system, involving X- and Y-specific regions, is ancient and predates species divergence. Therefore, we calculated the distribution of synonymous site divergence ($d_S$) between the putative orthologous genes located in X- and Y-specific regions to test whether the X–Y differentiation is relatively ancient or recent compared with species divergence within the genus. We used all predicted genes (23,200 genes) with complete coding sequences from *D. tokoro* Kita1 haplotype 1 assembly as the reference, and obtained (1) 76,090 orthologous or paralogous gene pairs by comparison with the Waka1 haplotype 1 assembly of *D. tokoro*, (2) 61,220 gene pairs by comparison with *D. alata* genome, (3) 75,576 gene pairs by comparison with *D. rotundata* genome. The $d_S$ values between possible orthologous gene pairs located on X- and Y-specific regions (XY gametologs) were compared to the three estimates of $d_S$ values: (1) *D. tokoro* intraspecific $d_S$ values; $d_S$ values of all orthologous and paralogous gene pairs between the Kita1 haplotype 1 reference genome and the Waka1 haplotype 1 reference genome, (2) interspecific $d_S$ of *D. tokoro–D. rotundata,* (3) interspecific $d_S$ of *D. tokoro–D. alata* (S25A Fig). The median $d_S$ value of 22 XY gametologs was 0.019 (S25B and S25C Fig), which is comparable to the median of *D.*

*tokoro* intraspecific $d_S$ values, 0.015. In contrast, it is > 20-fold lower than the median divergence between *D. tokoro* and *D. rotunada* (median of $d_S$ value: 0.49) or *D. tokoro* and *D. alata* (median of $d_S$ value: 0.48), (S25A Fig), suggesting that the divergence time between Y- and X- located genes are not as old as the divergence time between the *D. tokoro* and two other *Dioscorea* species.

In X–Y systems, divergence between homologous X- and Y-linked genes typically increases with distance from PARs, forming a stepwise pattern referred to as evolutionary strata. In *D. tokoro*, we also examined the presence of evolutionary strata, which are observed as groups of X–Y gene pairs showing distinct levels of synonymous divergence. We applied a threshold of $d_S < 0.5$ to filter out potential paralogous X–Y gene pairs with excessively large $d_S$ values, based on the maximum $d_S$ values observed in SDRs in previous studies [22,51]. We also evaluated it with a less stringent threshold of $d_S < 1.0$. The distribution of $d_S$ values of X–Y gene pairs in *D. tokoro* showed no clear evolutionary strata throughout the X and Y chromosomes (S26 Fig); we could not detect any signature of successive extension of recombination suppression following the X–Y differentiation in *D. tokoro*.

Finally, we conducted an interspecific comparison of SDRs among the *Dioscorea* species (Fig 4). In the genus *Dioscorea,* multiple sex-determination systems, including XY/XX and ZZ/ZW systems, have been reported and the different systems are scattered across its phylogeny (Fig 4A). We performed synteny analysis between *D. tokoro* and two other *Dioscorea* species: *D. alata*, which has an XY/XX system, and *D. rotundata*, which has a ZZ/ZW system. The 20–30 Mb region of sex chromosomes (chromosome 3) of *D. tokoro*, including the SDRs, were syntenic with the sex chromosome (chromosome 6; [35,44]) and chromosome 13 of *D. alata* (Fig 4B). In contrast, the SDR of *D. rotundata* is located in the terminal region of its sex chromosome (chromosome 11; [27]), and the sex chromosome shows no collinearity with the sex chromosomes of either *D. tokoro* or *D. alata* (Fig 4B). These results suggest transitions of the sex-determination system and SDRs within the genus *Dioscorea* (Fig 4C).

## The Y-specific genes *BLH9* and *HSP90* are specifically expressed during early male flower development

The male heterogametic sex determination (XY) in *D. tokoro* suggests that Y-specific genes or miRNAs function in sex determination. To identify the candidate genes or miRNAs involved in sex determination, we conducted transcriptome analysis via RNA-seq and small RNA-seq, given that small RNAs have also been reported to play roles in sex determination [12,74]. In *D. tokoro*, male and female flowers show sex-specific phenotypes after the bud stage (Fig 5A and 5B), indicating that the determination of sex phenotypes takes place before this stage. Therefore, we obtained RNA-seq and small RNA-seq data from male and female flowers at five stages of development as well as non-reproductive organs (Fig 5C and 5D). We mapped the RNA-seq reads to the Kita1 (male) haplotype 1 assembly, which includes the Y chromosome.

The Y-specific regions contained 56 potential protein-coding genes (Fig 5E and Table 1) and one primary miRNA (No. 20 in S4 Table). Of these, 14 genes (genes 1–14 in Fig 5E) were Y-specific and more highly expressed during the three stages of early male vs. female flower development based on the hypothesis test for differential expression of the RNA-seq data using negative binomial generalized linear models ($p$-value < 0.05), whereas no primary miRNA showed differential expression between male and female flowers. A sequence similarity search with BLASTX revealed that the 14 genes included ten genes encoding hypothetical proteins without functional information and four genes encoding proteins with functional information (Table 1).

We also compared the expression levels of genes between male and female flowers based on the $\log_2$ ratio of the mean of normalized counts ($\log_2$[fold change], $\log_2$FC). Within the 14 genes identified by differential expression analysis described above, we identified ten genes (genes 1, 4–10, 12, and 13 in Fig 5E) that showed more than 4-fold higher expression ($\log_2$FC > 2) during the early stages of male vs. female flower development. The ten genes included two genes encoding proteins with functional annotations: *A. thaliana BEL1-LIKE HOMEODOMAIN PROTEIN 9* (*AtBLH9*) and rice (*Oryza sativa*) *HEAT SHOCK PROTEIN 81–1* (*HSP81–1*). The eight remaining genes encode hypothetical proteins without functional information.

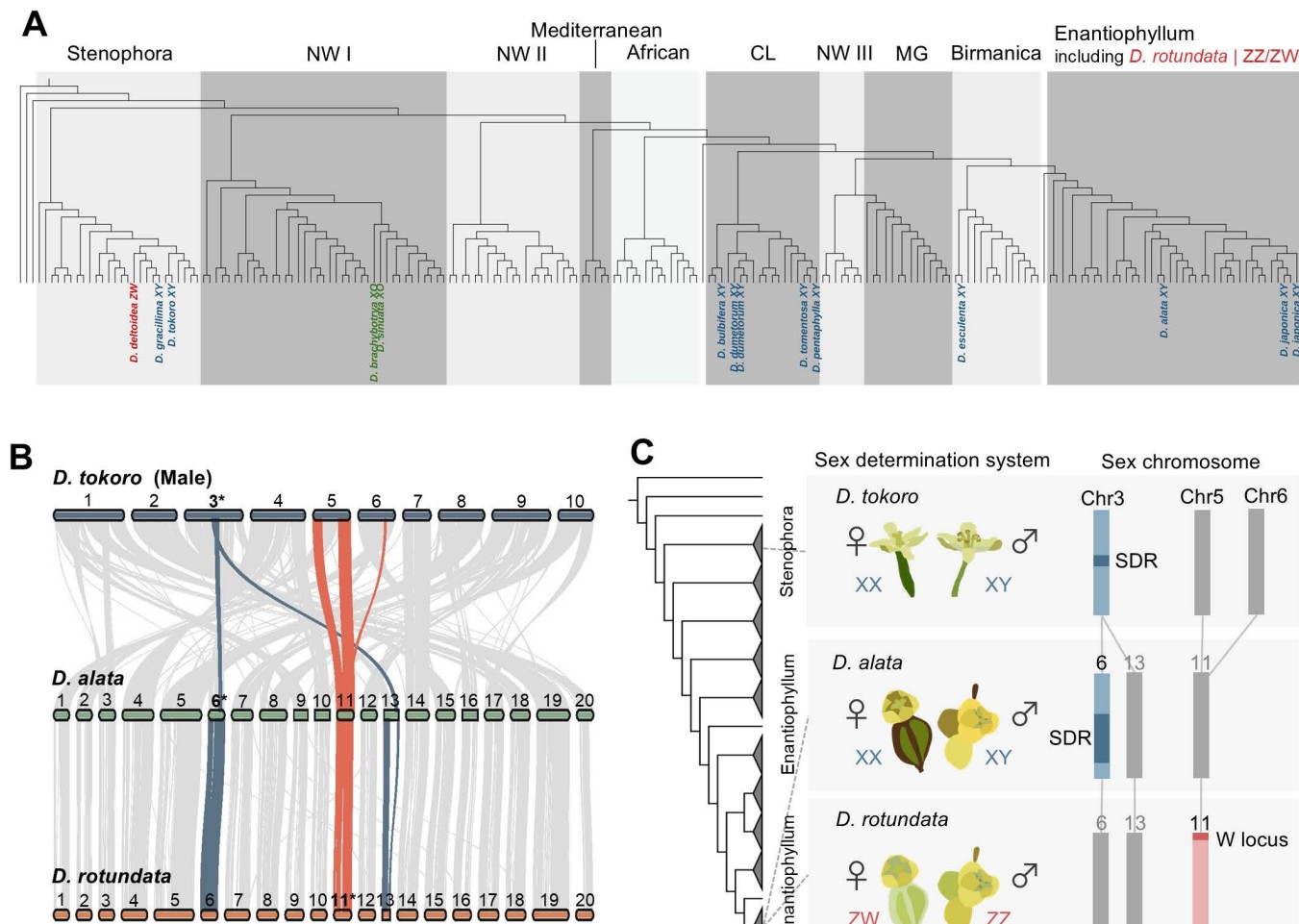

**Fig 4. Possible transitions of sex chromosomes in the genus *Dioscorea*. (A)** Distributions of sex determination systems on a phylogenetic tree of 183 species of *Dioscorea*. Each of the sections Stenophora and Enantiophyllum contains species with XY/XX and ZZ/ZW systems. The phylogenetic tree was adopted from [41]. *D. rotundata* was not included in the phylogenetic analysis of *Dioscorea* in [41], but it belongs to the section Enantiophyllum [147]. **(B)** Gene order–based synteny analysis of chromosomes of *D. tokoro* (male haplotype 1), *D. alata* [30] and *D. rotundata* [29]. Blue-colored bands between *D. tokoro* and *D. alata* indicate synteny blocks spanning 20-30 Mb of chromosome 3 (male haplotype 1) in *D. tokoro*, including the SDRs, and their corresponding regions in *D. alata*. Blue-colored bands between *D. alata* and *D. rotundata* indicate synteny blocks of chromosomes 6 and 13 in *D. alata* and their corresponding regions in *D. rotundata*. Red-colored bands indicate synteny blocks of chromosome 11 in *D. rotundata* and the corresponding regions in two other species. **(C)** Comparative schematic representation of sex chromosomes in three *Dioscorea* species.

When we compared the expression levels of the ten genes on the Y-specific region between male flowers and non-reproductive organs, only two genes (genes 1 and 5 in Fig 5E) had higher expression in male flowers than in non-reproductive organs ($p$-value < 0.05), and both these genes showed significantly higher expression in male flowers based on normalized counts ($\log_2 FC > 2$). Thus, we identified two Y chromosome–specific genes showing significantly higher expression in male flowers during the three stages of early development ($p$-value < 0.05 and $\log_2 FC > 2$) compared to female flowers and non-reproductive organs. The first gene, named *BLH9*, is a male-specific gene covered only by male-specific short-read sequences (Fig 5F) that is upregulated in male flowers during the three stages of early development (Fig 5G). A sequence similarity search by BLASTX showed that *BLH9* is closely related to the *A. thaliana* homeobox gene *AtBLH9*, which belongs to the *BELL* family within the TALE (three-amino-loop-extension) superfamily (S5 Table).

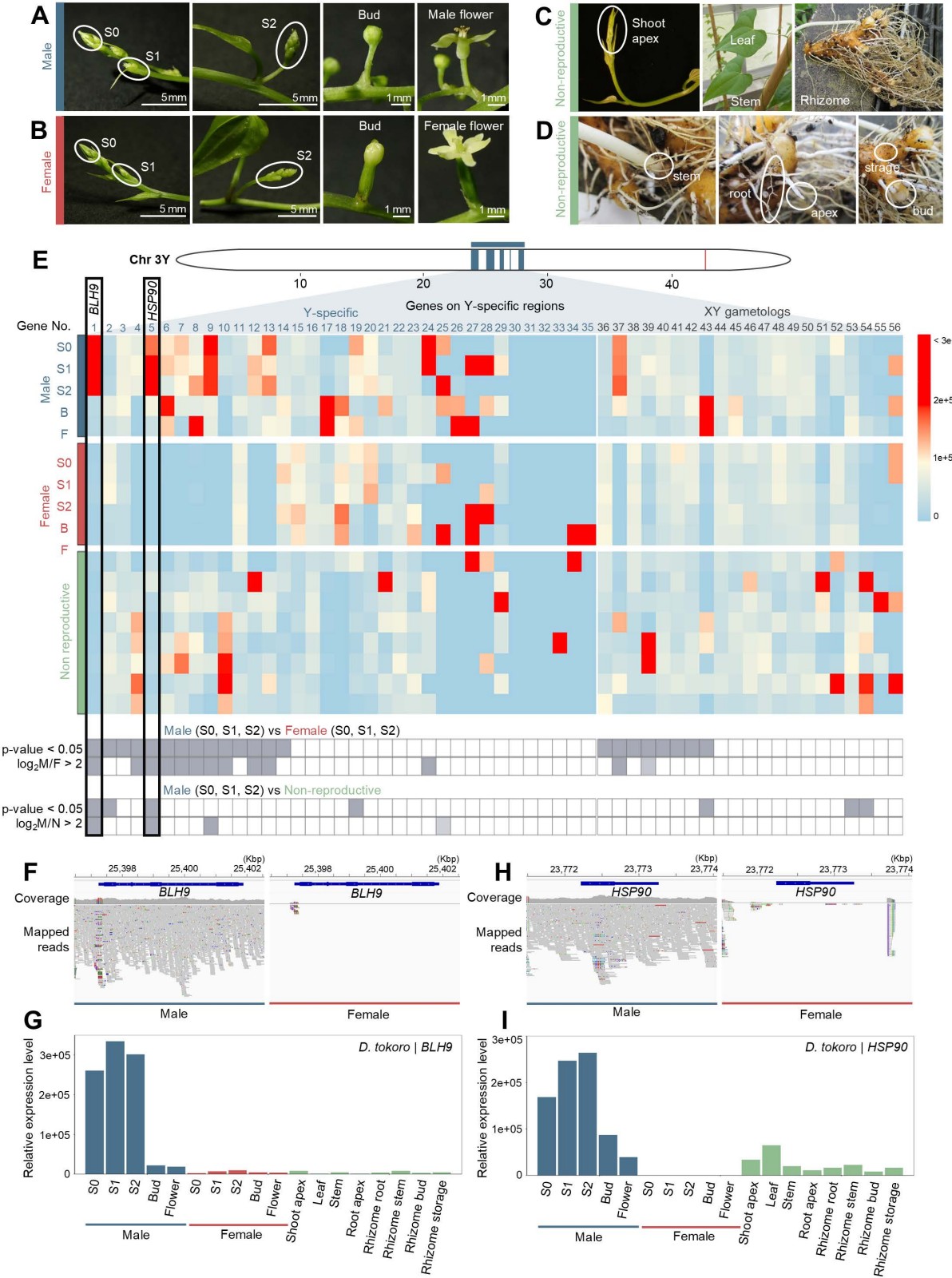

**Fig 5. The candidate genes *BLH9* and *HSP90* are located on Y-specific regions and are expressed at high levels during early male flower development. (A)** Five stages of male flower development used for RNA-seq and small RNA-seq analyses. **(B)** Five stages of female flower

development. **(C)** Non-reproductive parts of the plant, including the shoot apex, leaf, stem, and rhizome. The shoot apex is indicated with ellipses. **(D)** Parts of the rhizome, including the rhizome stem, apex, root, bud, and storage region. Each rhizome region is indicated with ellipses. **(E)** Heatmap of the expression levels of genes in the Y-specific region. The boxes under the heatmap show the DESeq2 results for each gene. The gray boxes indicate $p$-value < 0.05 based on differential expression testing by negative binomial generalized linear model or $\log_2 FC > 2$ based on the $\log_2$ ratio of the mean of normalized counts in each group in two comparisons: male (S0, S1, S2) vs. female (S0, S1, S2) and male (S0, S1, S2) vs. non-reproductive organs. *BLH9* and *HSP90*, highlighted by rectangles in the heatmap, are highly expressed during early stages of male flower development. **(F)** Coverage of short reads from male and female individuals on *BLH9*. *BLH9* was covered only by male short reads. **(G)** Relative expression levels of *BLH9* in male and female flowers at five stages of development and in non-reproductive organs. **(H)** Coverage of short reads on *HSP90* from male and female individuals. *HSP90* was covered only by male short reads. **(I)** Relative expression levels of *HSP90*. All expression levels are shown as normalized transcripts per million (TPM) values.

In the phylogenetic tree including all putative homologues of the TALE superfamily of *D. tokoro*, *BLH9* was placed in the same clade as *AtBLH9* together with a *D. tokoro* gene, *DtKita1h1 03399.t1.p1,* located on chromosome 1 (S27 Fig). The second gene, *HSP90*, is also male specific (Fig 5H) and was upregulated in male flowers during the three stages of early development (Fig 5I). *HSP90* shares similarity with the *A. thaliana* chaperone gene *HEAT SHOCK PROTEIN 90–4* (*AtHSP90.4*) (S5 Table). In the phylogenetic tree including all putative homologues of the *HSP90* protein family of *D. tokoro* and *A. thaliana*, *HSP90* showed a similarity to *AtHSP90.1–AtHSP90.4*, but its DNA sequence appeared to have diverged from other HSP genes of *D. tokoro* (S28 Fig). The male specificity of both genes was confirmed by PCR amplification of the DNA fragment using gene-specific primers and DNA templates from individuals of two natural *D. tokoro* populations: the HNMK population (northern Japan) and the KMMT population (southern Japan) (S29 Fig). In summary, The Y-specific genes *BLH9* and *HSP90*, specifically expressed during early male flower development, were identified as the primary candidate sex-determining genes.

### *BLH9* shares similar functions with *AtBLH9* and suppresses fruit development in *A. thaliana*

A genetic transformation method for *D. tokoro* has not yet been established. Among plants in the genus *Dioscorea*, successful transformation has been reported only in *D. rotundata* [75], which belongs to a section (Sect. Enantiophyllum) distant from *D. tokoro* (Sect. Stenophora) and possesses a ZZ/ZW sex-determination system [27]. Therefore, it is difficult to validate the functions of *D. tokoro* Y-specific genes by transformation of *Dioscorea* species.

To explore the biological role of the candidate gene *BLH9*, we overexpressed this gene and *AtBLH9* in *A. thaliana*. Phylogenetic analysis of *D. tokoro* BLH9 and putative homologues of BLH9 from monocotyledonous species (*Asparagus officinalis, Dioscorea alata, D. rotundata, Oryza sativa* and *Phoenix dactylifera*) as well as *A. thaliana* TALE (three-amino-loop-extension) superfamily proteins showed that AtBLH9 shared the greatest amino acid sequence similarity with BLH9 and its putative homologues (Fig 6A). AtBLH9 is involved in inflorescence architecture and fruit development, as overexpression or knockout of *AtBLH9* led to abnormal inflorescence development, including reduced inflorescence height and shorter fruit length, as well as irregular internode elongation, with extremely short and long internodes, respectively [76–81]. To compare the biological roles of *AtBLH9* and *BLH9*, we overexpressed *AtBLH9* and *BLH9* in *A. thaliana* plants driven by the CaMV35S promoter. For phenotypic evaluation, we used $T_2$ plants transformed with a construct expressing *BLH9* or *AtBLH9*. Wild-type *A. thaliana* Col-0 plants were used as a control.

We evaluated the relative expression levels of *BLH9* and *AtBLH9* in all $T_2$ plants by RT-qPCR using the $2^{-\Delta\Delta CT}$ method (S30 Fig). The $T_2$ plants were categorized into low-expressing ($2^{-\Delta\Delta CT} \leq 2$; indicated as Cont.) and overexpressing ($2^{-\Delta\Delta CT} > 2$; indicated as *OX*) groups. Compared to Col-0 plants and the low-expressing $T_2$ plants (*BLH9* Cont. and *AtBLH9* Cont.), the $T_2$ overexpressing plants (*BLH9 OX* and *AtBLH9 OX*) showed shorter average main inflorescence height, but the differences between *BLH9 OX* and Col-0 plants and that between *BLH9 OX* and *BLH9* Cont. were not statistically significant (Fig 6B and 6C). *BLH9 OX* had shorter fruits than the Col-0 plants and the *BLH9* Cont. (Fig 6D, 6E, and 6F). We also evaluated irregular internode elongation by examining variation in internode length in the main inflorescence. A large

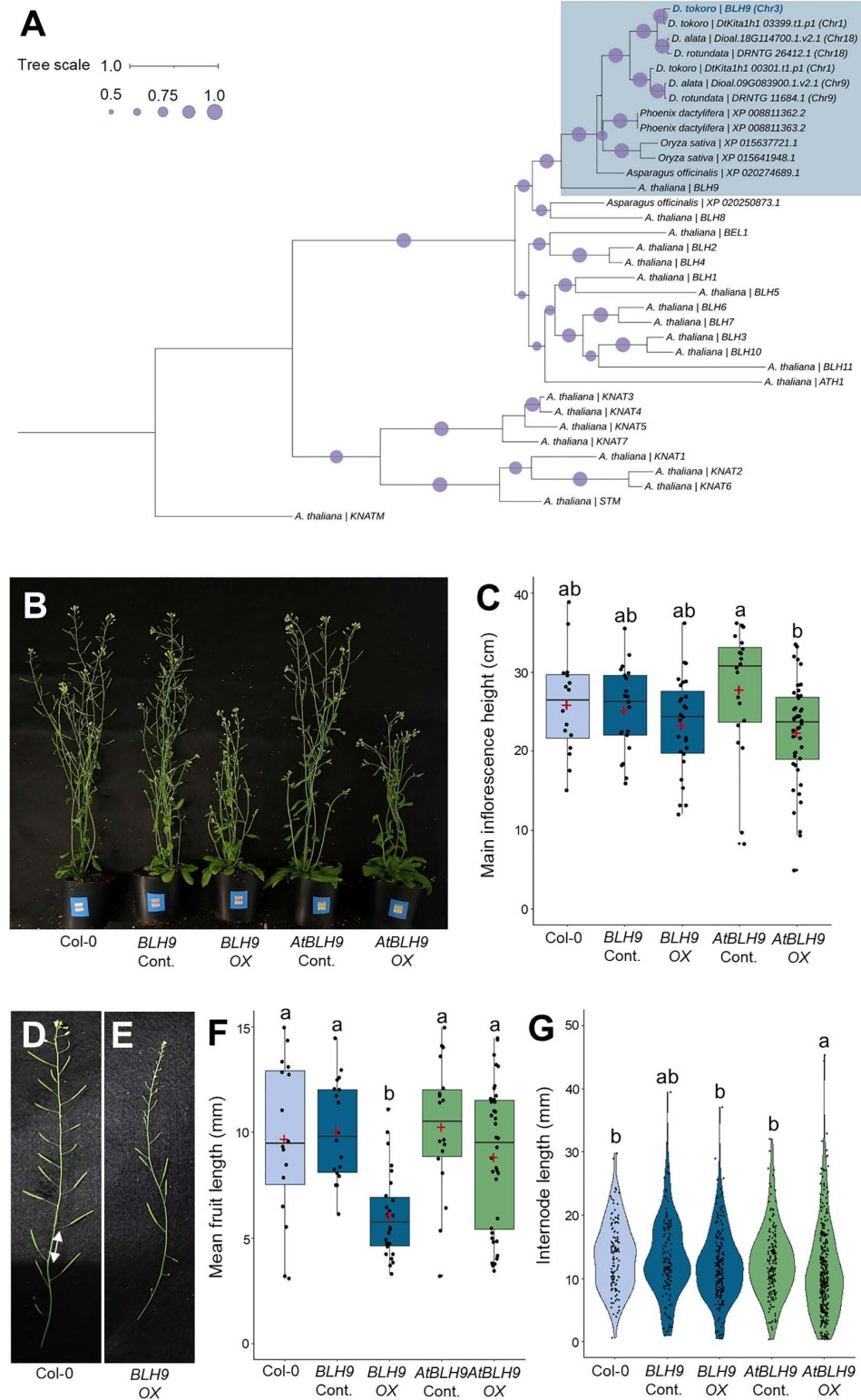

**Fig 6. The *D. tokoro* Y-specific gene *BLH9* plays similar roles to *AtBLH9* in inflorescence development. (A)** Phylogenetic tree of *BLH9* from *D. tokoro*, putative homologues of *BLH9* from monocotyledonous species (*Asparagus officinalis, Dioscorea alata, Dioscorea rotundata, Oryza sativa* and *Phoenix dactylifera*), and TALE superfamily proteins from *A. thaliana*. The circles in the tree nodes indicate bootstrap values. Tree scale indicates

amino acid substitutions per site. **(B)** Whole-plant phenotypes of Col-0, control (Cont.) and overexpression (*OX*) lines of *BLH9*, and control (Cont.) and overexpression (*OX*) lines of *AtBLH9*. Relative expression levels were calculated using the $2^{-\Delta\Delta CT}$ method, with gene expression in the overexpression lines (*OX*) defined as $2^{-\Delta\Delta CT} > 2$ and low expressing lines defined as $2^{-\Delta\Delta CT} \le 2$ (S30 Fig). The low expressing lines were used as control (Cont.). **(C)** Main inflorescence heights of the five lines. Red plus symbol, average; center line, median; box limits, upper and lower quartiles; whiskers, 1.5x interquartile range; points, each measurement. **(D)** Inflorescence of Col-0. White arrow indicates an internode. **(E)** Inflorescence of the *BLH9* overexpression line. **(F)** Mean fruit lengths of the five lines. **(G)** Variation in internode length of the five lines.

variation in internode length was observed in *AtBLH9 OX* (Fig 6G) but not in *BLH9 OX.* These results suggest that *BLH9* from *D. tokoro* shares some but not all functions with *AtBLH9* from *A. thaliana* and may be involved in suppression of fruit development.

## Discussion

In this study, we constructed haplotype-resolved genome assemblies and used them to explore the sex-determination mechanism of the wild yam *D. tokoro.* We found that this species has a male heterogametic sex-determination system (XY = male, XX = female) and that the sex-determination locus is situated on chromosome 3*.* This chromosome is differentiated into X- and Y-specific regions, and within the Y-specific region, we identified *BLH9* and *HSP90* as the primary candidate sex-determining genes*.*

*Dioscorea tokoro* contains homomorphic sex chromosomes without microscopically detectable size differentiation [34], suggesting this species might be in an early stage of sex chromosome evolution [6]. To investigate the initial differentiation of sex chromosomes, we constructed haplotype-resolved reference genomes. The haplotype 1 assemblies from the male individual (Kita1) and female individual (Waka1) contained six and two telomere-to-telomere contig assemblies, respectively. Compared to the haplotype 1 assemblies, the haplotype 2 assemblies resulted in a smaller genome size and large number of contigs. This may be caused by the assembly program, PECAT, which prioritizes the assembly of haplotype 1 [57].

In *D. tokoro*, chromosome 3 is differentiated into X and Y chromosomes with small X- (~4.8 Mb) and Y- specific (~4.5 Mb) regions. These regions are located in the middle of chromosome 3, in a region characterized by lower gene density and higher density of repetitive DNA sequences, represented by Ty3 LTRs, and may correspond to the pericentromeric region with low recombination [82–86]. Centromeres and pericentromeric regions are known to have suppressed recombination [87] and contain a higher density of retrotransposons and transposons [83,86]; particularly high levels of Ty3 LTRs are reported in multiple monocot species [82,84,85]. These results suggest that the initial SDR of *D. tokoro* occurred in a pericentromeric region with suppressed recombination, a pattern reminiscent of several dioecious species: white campion *Silene latifolia* [52], heartwing sorrel *Rumex hastatulus* [88], garden asparagus [89], the poplar *Populus tremuloides* [90], and the papaya relative *Vasconcellea parviflora* [91]. Moreover, recent studies suggested that sex chromosome formation may occur in regions with pre-existing recombination suppression [92], raising the possibility that the establishment of SDR in *D. tokoro* was also influenced by such pre-existing recombination suppression.

An intriguing aspect of SDRs in the genus *Dioscorea* is that multiple sex-determination systems, including XY/XX and ZZ/ZW systems, have been reported, and the different systems are scattered across its phylogeny. *Dioscorea alata*, a member of section Enantiophyllum, the most derived group within the genus *Dioscorea*, contains a large pericentric inversion in its sex chromosome that includes an ~7.6-Mb SDR, accounting for ~44% of the sex chromosome [44]. The sex chromosome of *D. alata* is collinear with half of the sex chromosome of *D. tokoro*, which belongs to section Stenophora, the most ancestral section in the genus [44]. The genomic region around the SDR of *D. tokoro* is also syntenic with the sex chromosome of *D. alata* in our synteny analysis. However, $d_S$ analysis suggested that the divergence between *D. tokoro* Y- and X-linked genes is more recent than the divergence between *D. tokoro* and *D. alata*. These findings suggest a possibility of convergent evolution in which the same genomic regions came to be involved in sex determination

independently in *D. tokoro* and *D. alata*. The small value of $d_S$ in *D. tokoro* X- and Y-gametologs suggests that current *D. tokoro* sex determination system is relatively new, possibly originated by a turnover from the original system present in the ancestral *Disocorea* species. In contrast, the SDR of *D. rotundata* is located in the terminal region of chromosome 11 which shows no collinearity with the sex chromosomes of *D. tokoro* or *D. alata* [27,44]. These results point to transitions of the sex-determination system and SDRs within a single genus. Transitions in the heterogametic systems of sex chromosomes, such as shifts from XY/XX to ZZ/ZW systems, have been investigated in only a few plant groups, the genus *Populus* [93] and *Salix* [94], where SDR structures have been influenced by genome duplication events and/or the spread of transposable elements. The organismal sex determination systems are usually ancient and highly conserved; sex chromosomes have evolved and persisted over more than 100 million years in groups as diverse as therian mammals, birds, lizards, and stick insects [95]. However, flexible transitions of heterogametic systems have been reported in a limited cases including frogs [96], African cichlids [97], and a few plant species. Further studies in the genus *Dioscorea* may provide new insights into the evolutionary mechanisms underlying such flexible transitions of sex chromosomes and SDRs.

Despite the previous studies of sex-determination systems in the genus *Dioscorea*, the molecular basis for its sex determination has not been reported. In this study, we identified two Y-specific genes, *BLH9* and *HSP90*, as candidate sex-determining genes related to the regulation of female and male floral organ development. The developmental program leading to carpel initiation is induced by one of the floral homeotic genes, *AGAMOUS* (*AG*) [98]. The newly identified gene *BLH9* shares sequence similarity with *AtBLH9* of *A. thaliana*, which functions as a repressor of *AG* together with *LEUNIG* and *SEUSS* [99]. *AtBLH9* plays multiple roles in inflorescence architecture and fruit development [76–80]. *AtBLH9*-overexpressing plants have reduced inflorescence height and irregular inflorescence architecture [81]. We also observed reduced inflorescence height and irregular internode elongation in *A. thaliana AtBLH9* overexpression lines. The short fruit length phenotype occurred in the overexpression lines of *BLH9.* Therefore, similar to AtBLH9, the *Y*-encoded BLH9 of *D. tokoro* might function as a repressor of *AG* and suppress female development in *D. tokoro*. Future research should aim to establish functional validation of BLH9 in *D. tokoro* as well as to identify the downstream mechanism of *BLH9* expression that controls dioecious floral formation. Another candidate gene, *HSP90*, shares sequence similarity with the *A. thaliana* chaperone gene *AtHSP90.4.* In *A. thaliana,* AtHSP90.1 to AtHSP90.4 function in the nucleus and cytoplasm and share high sequence similarity [100]. AtHSP90 forms a complex with BRASSINOSTEROID INSENSITIVE 1-EMS-SUPRESSOR 1 (BES1) and BRASSINAZOLE RESISTANT 1 (BZR1) in the nucleus and cytoplasm [101–103], which is required for their downstream function [100]. BZR proteins, including BES1 and BZR1, are indispensable for pollen production; the simultaneous knockout of *BES1*, *BZR1*, and their homologs resulted in male sterility [104]. The *Y*-encoded gene *HSP90* of *D. tokoro* might also be required for successful male flower development.

The two primary candidate genes for sex determination in *D. tokoro*, described above, may be consistent with a major hypothesis for the evolution of dioecy: two sequential mutations affecting male and female fertility are supposed to have led to the evolution of separate sexes controlled by the male-promoting sex determinant (M factor) and the suppressor of female development (SuF) [105–107]. This two-factor model is also supported by studies in multiple dioecious plants, including garden asparagus, kiwifruit (*Actinidia chinensis*) and date palm (the genus *Phoenix*) [108]. In garden asparagus, *SOFF* and *aspTDF1* on the Y chromosome independently act to suppress pistil development and promote anther development, respectively [16–19]. Kiwifruit also follows a two-gene system for sex determination: *Shy Girl* and *Friendly boy* independently suppress feminization and promote male sex determination in tapetal cells, respectively [109,110]. These cases support the hypothesis of the evolution of dioecy in two-factor model, the putative SuF genes were established by gene duplication/translocation on the proto-Y chromosome, contributing to the formation of Y chromosome. Conversely, the putative M genes were lost from the proto-X chromosome, leading to the establishment of X chromosome [110]. The candidate *D. tokoro* genes, *BLH9* and *HSP90*, may be involved in the suppression of female function and the establishment of male function, respectively. Phylogenetic analysis of putative homologues of BLH proteins in *D. tokoro* suggested that *BLH9* may have originated through duplication of a gene located on chromosome 1, whereas *HSP90* showed no

close sequence similarity to genes on autosomal chromosomes. These results suggest that the putative dominant SuF gene, *BLH9* in *D. tokoro*, may have been established by gene duplication, whereas the putative M gene, *HSP90* in *D. tokoro*, may have been lost from the X chromosome. The candidate *D. tokoro* genes identified in this study belong to different classes of genes previously identified in dioecious plants, suggesting that mutations in various genes related to flower development led to the evolution of dioecy.

An additional intriguing question concerns how reproductive organs and sexual dimorphism are established after sex determination occurred, a process that is closely related to the emergence of dioecy and the evolution of SDRs [111,112]. Key genomic regions and genes for sex determination have been identified in various plant species in recent years; however, the downstream signaling mechanisms resulting in the differential development of female and male floral organs remain elusive [113]. The Y-specific region in *D. tokoro* identified in this study includes genes that are upregulated in male flowers during late development, e.g., gene no. 43 in Fig 5E. This gene shares sequence similarity with *A. thaliana EXO70C1*, encoding a component of the exocyst complex (Table 1). *EXO70C1* is expressed in pollen and functions in pollen tube growth [114]. Following sex determination conferred by *BLH9* and *HSP90* expression, Y-specific downstream genes such as *EXO70C1* might function in the completion of successful male fertility. The Y-specific region of *D. tokoro* also includes genes that are highly expressed in non-reproductive organs; these genes might be involved in the development of as-yet-unrecognized male-specific characteristics. Further study of these Y- or X-specific genes should help reveal the downstream signaling mechanisms that function after sex determination and play roles in the evolution of sex chromosomes in dioecious plants.

## Conclusion

In summary, the reconstructed X- and Y-specific regions of chromosome 3 in *D. tokoro* suggest that the SDR of this species is likely present in the pericentromeric region of chromosome 3. We identified the Y-encoded genes *BLH9* and *HSP90* located in the SDR as prime candidates for genetic components underlying sex determination in *D. tokoro*. These two candidate genes are highly expressed during the early stages of male flower development and are predicted to suppress femaleness and promote maleness. Further molecular analyses of genetic components for sex determination in other *Dioscorea* species will shed light on the complex evolution of dioecy in this genus.

## Materials and methods

### Plant materials

To construct the reference genome, a female *D. tokoro* plant (Waka1; original code: DT49) collected from Tahara, Wakayama Pref., Japan (33°32'16.8"N, 135°51'36.0"E) was crossed with a male *D. tokoro* plant (Kita1; original code: 110628-5) collected from Waga-Sennin, Kitakami, Iwate Pref., Japan (39°17'42.0"N, 140°53'45.6"E). The 186 $F_1$ progeny comprised 38 female plants, 89 male plants, and 59 non-flowering plants. To identify sex-linked regions, female and male individuals were collected from three wild populations: a population in northern Japan (KTKM; Kitakami, Iwate Pref., Japan; 39°18'25.0"N, 140°54'07.0"E); a population in central Japan (SHG; Koka, Shiga Pref., Japan; 34°56'24.1"N, 136°13'05.3"E); and a population in southern Japan (FKOK; Kasuya, Fukuoka Pref., Japan; 33°38'08.7"N, 130°30'38.2"E). For SDRs confirmation, five females and five males were collected from a wild population (KZGW; Kozagawa, Wakayama Pref., Japan; 33°32'24.1"N, 135°46'29.3"E). For transcriptome analysis, RNA-seq data were collected from 18 tissue samples. Male and female flowers were collected from wild populations in Takizawa and Kitakami, Iwate Pref., Japan, and non-reproductive organs were collected from Kita1. For small RNA-seq, 15 samples were collected from the wild population in Koka, Shiga Pref., Japan. To check the male specificity of the candidate genes, five females and five males were collected from two wild populations: one in northern Japan (HNMK; Hanamaki, Iwate Pref., Japan; 39°22'10.0"N, 141°09'16.0"E) and one in southern Japan (KMMT; Kumamoto, Kumamoto Pref., Japan; 32°53'34.8"N, 130°39'22.7"E).

## DNA/RNA extraction, library preparation, and sequencing

For Oxford Nanopore Technologies (ONT) sequencing, genomic DNA was extracted from fresh leaves of Waka1 (female) and Kita1 (male) using NucleoBond HMW DNA (Macherey-Nagel, Düren, Germany). The DNA was subjected to size selection and purification with Short Read Eliminator XL (Circulomics, Baltimore, MD, USA). Long-read libraries were constructed using a Ligation Sequencing Kit V14 (SQK-LSK114). The libraries were sequenced on R10.4.1 flow cells (FLO-PRO114M) using a PromethION 2 Solo device. The raw sequencing data were subjected to base calling using dorado v0.8.1.

For Illumina sequencing, genomic DNA was extracted from Waka1 (female) and Kita1 (male) using a NucleoSpin Plant II Kit (Macherey-Nagel). Libraries for Waka1 (female) were constructed using a Collibri ES DNA Library Prep Kit for Illumina Systems (Invitrogen, Camarillo, CA, USA) and a TruSeq DNA PCR-Free LT Library Prep Kit (Illumina, San Diego, CA, USA). Libraries for Kita1 (male) were constructed using a TruSeq DNA PCR-Free LT Library Prep Kit. The libraries were sequenced using the MiSeq and HiSeq X systems. Genomic DNA was extracted from female and male individuals from northern (KTKM), central (SHG), and southern (FKOK) Japan using a DNeasy Plant Maxi Kit (Qiagen, Hilden, Germany). The DNA was purified by phenol/chloroform extraction and ethanol precipitation. Sequencing libraries were constructed using a Collibri ES DNA Library Prep Kit for Illumina Systems with a fragment length of ~350 bp and sequenced using the HiSeq X system (150 bp paired-end reads). Genomic DNA was extracted from five females and five males from KZGW population using Maxwell RSC Plant DNA Kit (Promega, Madison, WI, USA). Sequencing libraries were constructed using a Collibri PCR-free ES DNA Library Prep Kit for Illumina Systems with a fragment length of ~350 bp and sequenced using the Novaseq X Plus system (150 bp paired-end reads).

RAD-seq was performed as previously described [27]. Genomic DNA was extracted from fresh leaves of Waka1, Kita1, and 186 $F_1$ individuals using a NucleoSpin Plant II Kit (Macherey-Nagel). The DNA was digested with the restriction enzymes PacI and NlaIII and used to prepare libraries, and 75-bp paired-end reads were sequenced on the Illumina NextSeq 500 platform. Adapters and unpaired reads were removed using FaQCs and PRINSEQ lite. The filtered RAD-seq reads were used for linkage mapping and association analysis (S1 Data).

The RNA-seq data were obtained from 18 samples, including male and female flowers and non-reproductive organs of *D. tokoro.* The samples included male and female flowers at five stages of development: inflorescence stage 0, inflorescence stage 1, inflorescence stage 2, buds, and flowers. The samples also included eight non-reproductive organs: the vegetative shoot apex, leaf, stem, root apex, rhizome root, rhizome stem, rhizome bud, and rhizome storage tissue. Total RNA was extracted from the samples using an RNeasy Plant Mini Kit (Qiagen). The cDNA library was constructed using a TruSeq RNA Sample Prep Kit V2 (Illumina) and sequenced on the Illumina NextSeq 500 platform.

Small RNA-seq data were obtained from 15 samples, including male and female flowers and non-reproductive organs of *D. tokoro.* The samples included male and female flowers at five stages of development: inflorescence stage 0, inflorescence stage 1, inflorescence stage 2, buds, and flowers. The samples also included three non-reproductive organs: the vegetative shoot apex, male and female leaves, and male and female stems. Total RNA was extracted from the samples using Ambion Plant RNA Isolation Aid (Ambion, Austin, TX, USA), and the small RNA fraction was isolated using a mirVana miRNA Isolation Kit (Ambion). The small RNA libraries were constructed using a NEBNext Multiplex Small RNA Library Prep Set for Illumina (New England BioLabs, Ipswich, MA, USA) and DNA Clean & Concentrator-5 (Zymo Research, CA, USA). The libraries were sequenced on the NovaSeq 6000 platform.

## Reference assembly

The genome size of *D. tokoro* Kita1 was estimated by flow cytometry using nuclei prepared from fresh leaf samples. *Dioscorea rotundata* accession TDr96-F1 (570 Mb) [27] was used as an internal reference standard. DNA from isolated nuclei was stained with propidium iodide and analyzed using a Cell Lab Quanta SC Flow Cytometer (Beckman Coulter, USA). The genome size of *D. tokoro* was estimated to be ~388 Mb (570 Mb × 0.68) (S2 Fig).

To construct the reference genome based on the predicted genome size, the ONT sequencing data for Waka1 (female) and Kita1 (male) were filtered using chopper v0.8.0 [115]. The filtered reads from Waka1 (female) were assembled using PECAT [57], resulting in 212 contigs with N50 of 19,154,004 bp and a total size of 425.1 Mb for haplotype 1, and 346 contigs with N50 of 3,042,808 bp and a total size of 342.2 Mb for haplotype 2. The filtered reads from Kita1 (male) were also assembled using PECAT [57], resulting in 128 contigs with N50 of 33,851,599 bp and a total size of 415.0 Mb for haplotype 1, and 415 contigs with N50 of 1,379,312 bp and a total size of 300.8 Mb for haplotype 2. The assemblies were polished using Racon v1.5.0 [116] and Medaka v1.7.2 (Oxford Nanopore Technologies, 2018). The assemblies were also polished twice with Hypo v1.0.3 [117] using Illumina short reads from Waka1 (female) and Kita1 (male) filtered using FaQCs v2.08 [118].

**TE and gene annotation**

To annotate repetitive sequences, *de novo* repeat libraries were constructed for each assembly using EDTA (The Extensive de novo TE Annotator) v2.1.0 [119]. Using the resulting files, each genome FASTA file was soft-masked with the perl script make_masked.pl provided in EDTA.

For transcriptome-based gene identification using RNA-seq data from 18 samples of *D. tokoro*, poly(A) sequences and reads shorter than 50 bp were removed from the raw RNA-seq reads with FaQCs v2.10 [118]. Low-quality bases on the ends of reads with an average quality score of <20 were trimmed using PRINSEQ lite v0.20.4 [120]. Low-quality reads with an average read quality score of <20 were also removed using PRINSEQ lite v0.20.4. To obtain transcriptomes, the filtered RNA-seq reads were aligned to the assembled contigs with HISAT2 v2.2.1 [121]. The transcriptomes were assembled using StringTie v3.0.0 [122], and open reading frame regions were identified using TransDecoder v5.5.0 [123]. Gene models were excluded from further analysis if their coding sequences (CDSs) contained an internal stop codon, if the CDS lengths were not multiples of three, or if they encoded proteins shorter than 50 amino acids.

BRAKER2 v2.1.6 [124] was used for *ab initio* gene prediction using the filtered RNA-seq data from *D. tokoro* and protein homology information for *D. alata* and *D. rotundata*. Protein sequences of *D. alata* TDa95/00328 and *D. rotundata* TDr96_F1 were downloaded from NCBI (accession number GCA_020875875.1 and GCF_009730915.1, respectively). The predicted genes were categorized into three groups: gene models fully supported by protein hints, gene models at least partially supported by protein hints, and gene models without any support using the python script selectSupportedSubsets.py in BRAKER. Gene models fully supported by protein hints were selected, and gene models in which the CDS contained an internal stop codon, was comprised of a number of nucleotides that was not a multiple of three, or encoded a protein of <50 amino acids were removed.

Finally, all predicted genes were merged using GffCompare v0.12.6 [125]. *Ab initio* predicted genes were selected with no or less overlapping with genes predicted by StringTie. Gene predictions obtained from GeneMark in BRAKER2 were excluded. The final annotations include 21,308 genes in the Waka1 (female) haplotype 1 assembly, 17,710 genes in the Waka1 (female) haplotype 2 assembly, 23,200 genes in the Kita1 (male) haplotype 1 assembly, and 18,145 genes in the Kita1 (male) haplotype 2 assembly. To evaluate the completeness of the gene sets in the final scaffolds, BUSCO (Bench-Marking Universal Single Copy) v5.5.5 [56] was used with "transcriptome" as the assessment mode and Embryophyta odb10 as the database (S2 Table).

**Generation of chromosomes using the pseudo-testcross approach**

The pseudo-testcross approach [58] was used to construct chromosomes from the assembled contigs. SNP-type heterozygous markers and PA-type heterozygous markers were obtained from RAD-seq data from Waka1, Kita1, and 186 F$_1$ individuals. Parental line–specific heterozygous markers were identified as previously described [29] with several modifications (S1 Text). Linkage maps were constructed based on the heterozygous SNP-type and heterozygous PA-type markers. For each marker set (female-parent-heterozygous and male-parent-heterozygous marker sets), the markers were converted to genotype-formatted data to construct genetic linkage maps using MSTmap v1.0 [126]. After performing

MSTmap, complemented-phased duplex linkage groups were generated by coupling- and repulsion-type markers under the pseudo-testcross approach. Finally, we pruned correlated flanking markers to remove redundant markers. After all filtering steps, two parental-specific linkage maps were constructed and visualized using R/qtl v.1.70 [60] and Asmap [127]. Based on the two parental-specific linkage maps, the contigs were anchored and linearly ordered as chromosomes using ALLMAPS [128]. Gene order–based synteny of the four constructed reference genomes was detected using MCscan [129]. Detailed methods are shown in S1 Text.

## Identification of sex-linked regions by association analysis and coverage analysis

To identify sex-determination systems and sex-linked regions, association analysis was conducted using RAD-seq data from the 127 flowering $F_1$ progeny, which segregated into 38 females and 89 males. The RAD-seq data were aligned to the newly constructed female and male chromosomes using BWA v0.7.18-r1243-dirty [130]. SNP-type markers and PA-type markers were identified based on these alignments (S1 Text).

The associations between the genotypes and sex phenotypes of the 127 flowering $F_1$ individuals were calculated using Fisher's exact test. The $q$-value of Fisher's exact test was obtained for each marker by comparing the frequencies of particular alleles and sex phenotypes categorized as female or male. The false discovery rate (FDR) was set to 0.05 and was corrected by Benjamini-Hochberg correction. The log-transformed $q$-values ($-\log10(q)$) for each position were visualized as Manhattan plots. In addition, the associations between the genotypes and sex phenotypes were calculated using Fisher's exact test with equal numbers of randomly selected female and male $F_1$ individuals (38 females and 38 males). The associations between the genotypes and sex phenotypes were also checked by simple interval mapping using R/qtl v.1.70 [60]. The alignment-based dot plot between the chromosome 3 of Waka1 (female) haplotype 1 and Kita1 (male) haplotype 1 were generated using D-Genies [131]. Chromosome 3 of Kita1 (male) haplotype 1 was used as the target sequence, and chromosome 3 of Waka1 (female) haplotype 1 was used as the query sequence. The full plot was generated with noise filtering, and the inset panel was generated without noise filtering, using the D-Genies algorithm that removes small matches based on the alignment size and frequency.

The structural differences in chromosome 3 of Waka1 (female) haplotype 1 and Kita1 (male) haplotype 1 were detected by coverage analysis using Illumina short reads from four combinations of male and female individuals: male parent Kita1 and female parent Waka1 as well as male and female individuals collected from northern, central, and southern Japan. The Illumina short reads were filtered using FaQCs v2.10 [118] and aligned to each reference genome. The alignment to each haplotype reference was conducted using BWA v0.7.17-r1188 [130]. The alignment depths were calculated with Samtools v1.16.1 [132]. The alignment depths were normalized by dividing the values with the mean depth of all positions on the chromosomes. To reduce the effects of error on the alignment depths, a sliding window approach was employed (window size = 150 kb, step size = 10 kb). We identified regions in which the male depth was 0.5 (half of mean normalized depth) ± 0.15 and the female depth was 1 (mean normalized depth) ± 0.15, and regions in which the male depth was 0.5 (half the mean normalized depth) ± 0.15 and the female depth was < 0.15.

SDRs were also estimated using two previously reported methods, SDpop [63] and RADSex [64]. For SDpop analysis, Illumina short reads from five females and five males from KZGW population filtered using FaQCs v2.08 [118] and aligned to each reference genome: Waka1 (female) and Kita1 (male) haplotype 1. The alignment to each haplotype was conducted using BWA v0.7.17-r1188 [130]. After obtaining SNP-based genotypes using Bcftools v1.15 [132], posterior probability for segregation types were calculated by SDpop [63]. In this analysis, only the polymorphic sites located within coding regions of orthologous gene pairs of chromosome 3Y (from Kita1 haplotype 1) and chromosome 3X (from Waka1 haplotype 1) were used. The orthologous gene pairs ($<1 \times 10^{-50}$ in BLASTP analyses, max_target_seqs = 2) were obtained following [13,22,133]. For RADSex analysis, we used RAD-seq data from 127 flowering $F_1$ individuals. Based on the filtered RAD-seq data and sex phenotypes, probability of association with sex, $-\log10(p)$, were calculated by RADSex v1.2.0 [64].

Based on the results of our association analysis, alignment-based dot plot analysis, mapping coverage analysis and SDRs detection analysis, regions including depth differences within the 23–29 Mb interval on X chromosome and within the 20–30 Mb interval on Y chromosomes were defined as X and Y-specific regions, respectively. The start position (23,395,001 bp) to end position (28,209,000 bp) of regions, in which the male depth was 0.5 (half of mean normalized depth) ± 0.15 and the female depth was 1 (mean normalized depth) ± 0.15, in X chromosome were defined as X-specific regions. The start position (23,759,001 bp) to end position (28,256,000 bp) of regions, in which the male depth was 0.5 (half of mean normalized depth) ± 0.15 and the female depth was < 0.15, in Y chromosome were defined as Y-specific regions. In this step, start and end positions of the depth differences were detected using higher-resolution sliding windows (window size = 20 kb, step size = 1 kb).

## Genome structures of the X and Y chromosomes

Gene density and repetitive sequence accumulation values were obtained from gene annotation and the predicted repetitive sequences of the Waka1 (female) and the Kita1 (male) haplotype 1 assembly. Repetitive sequences included Copia LTR retrotransposons, Ty3 (gypsy) LTR retrotransposons, hAT TIR transposons, and Helitrons predicted by EDTA v2.1.0. Gene density and retrotransposon accumulation were visualized using the jcvi package [129].

Synonymous site divergence ($d_S$) between X- and Y-linked gametologs was obtained following [13,22,133] with slight modifications. Orthologous or paralogous gene pairs (<$1 \times 10^{-50}$ in BLASTP analyses, max_target_seqs = 2) obtained from three comparison: female individual (Waka1) and male individual (Kita1) of *D. tokoro*, male (Kita1) *D. tokoro* and *D. alata* [30], male (Kita1) *D. tokoro* and *D. rotundata* [29]. The gene pairs were aligned in codon frames using MAFFT v7.475 [134] with the option "--localpair --maxiterate 1000" and Pal2Nal v14 [135]. Based on the in-codon-frame alignments, we calculate the Jukes and Cantor corrected values of $d_S$ using KaKs_Calculator v2.0 [136]. Gene pairs located in the Y-specific regions of male *D. tokoro* and the X-specific regions of female *D. tokoro* were defined as XY gametologs. The gametologs with $d_S$ values > 1 may not be real orthologs but may represent paralogs, so these were removed from XY gametologs. To evaluate the $d_S$ values of all gene pairs across X and Y chromosomes, we applied a threshold of $d_S < 0.5$ based on the maximum $d_S$ values in the SDRs reported in the previous studies [22,51] since these gene pairs may represent paralogs, not orthologs (S26A and S26C Fig). We also applied a less stringent threshold of $d_S < 1.0$ (S26B and S26D Fig).

To compare the sex chromosomes of *D. tokoro* (male), *D. alata* [30] and *D. rotundata* [29], Gene order–based synteny of chromosomes was detected using MCscan [129]. The orthologous regions were identified using the "jcvi.compara.catalog ortholog" and the "jcvi.compara.synteny screen" function with at least 30 collinear gene blocks. The detected collinearity was visualized using the "jcvi.graphics.karyotype" function.

## Identification of candidate genes and miRNAs for sex determination

For transcriptome analysis of genes, the filtered RNA-seq data from 18 samples, including male flowers, female flowers, and non-reproductive organs, were aligned to the assembled contigs with HISAT2 v2.2.1 [121], and the aligned reads were counted using the featureCounts function in Subread v2.0.1 [137]. Differential expression analysis was performed with DESeq2 v3.15 [138] for two comparisons: male vs. female flowers at three stages of early development (inflorescence stages 0, 1, and 2) and male flowers at three stages of early development vs. non-reproductive organs. The FDR threshold was set to 0.05.

For transcriptome analysis of miRNAs, small RNA-seq data from 15 samples, including male flowers, female flowers, and non-reproductive organs, were filtered using FaQCs v2.10 [118] and Seqkit v2.3.0 [139]. Based on the small RNA-seq data, miRNAs were predicted using miRDeep-P2 v1.1.4 [140] as described previously [141]. The trimmed small RNA-seq reads were then aligned to the predicted miRNA references with Bowtie v1.3.1 [142]. The aligned reads were counted using the perl script bam2ref_counts.pl [141]. Differential expression analysis was performed with DESeq2 v3.15 [138]

to compare male vs. female flowers at three stages of early development (inflorescence stages 0, 1, and 2). The FDR threshold was set to 0.05.

In the first step of candidate gene and miRNA identification, 56 genes and one primary miRNA located on Y-specific regions were identified. In the second step, within the genes and miRNA on the Y-specific regions, 14 genes (and no miRNA) were detected that were Y-specific and highly expressed during these three stages of early male vs. female flower development ($q$-value < 0.05). Within the 14 genes, ten genes were identified that were significantly upregulated in male vs. female flowers during the same three stages of development ($\log_2$FC > 2). In the third step, within the ten genes, two genes were identified that were highly expressed in male flowers compared to non-reproductive organs during these three stages ($q$-value < 0.05). The two genes were significantly upregulated in male flowers compared to non-reproductive organs during the three stages ($\log_2$FC > 2). Finally, two Y-specific genes were selected with significantly upregulated expression in male flowers during the early three stages of development ($p$-value < 0.05 and $\log_2$FC > 2) in both transcriptome comparisons. The similarities of the candidate gene products were compared to known proteins using BLASTX with the Swiss-Prot function. The male-specific expression of the two candidate genes was confirmed by PCR amplification from five females and five males from two wild populations, one from northern Japan and one from southern Japan.

## Overexpressing *BLH9* and *AtBLH9* in *Arabidopsis thaliana*

The homologous genes of *BLH9* were subjected to phylogenetic analysis using protein sequences of BLH9, putative homologues of BLH9 from monocotyledonous species (*Asparagus officinalis, Dioscorea alara, D. rotundata, Oryza sativa* and *Phoenix dactylifera*) and TALE superfamily members in *A. thaliana.* The sequences were aligned using MAFFT v7.475 with the L-INS-i strategy [143]. Based on the alignment, a phylogenetic tree was constructed with the maximum likelihood method using iqtree v2.1.2 [144]. The tree was visualized using Interactive Tree Of Life (iTOL) v7.0 [145].

To overexpress *BLH9* and *AtBLH9* in *A. thaliana*, full-length cDNA clones were used for In-Fusion cloning. Total RNA was extracted from *D. tokoro* and *A. thaliana* using a Maxwell RSC Plant RNA Kit (Promega) and was reverse-transcribed into cDNA using a ReverTra Ace qPCR RT Kit FSQ-101 (TOYOBO, Osaka, Japan). *BLH9* fragment was amplified from the *D. tokoro* cDNA using KOD FX Neo (TOYOBO). *AtBLH9* was amplified from *A. thaliana* cDNA using PrimeSTAR GXL DNA Polymerase (Takara Bio). Each cDNA fragment was cloned into EcoRI and BamHI sites of the pBICP35 binary vector containing the CaMV35S promoter [146] using the In-Fusion HD Cloning Kit (Takara Bio). *pBICP35::BLH9* and *pBICP35::AtBLH9* were transformed into *Agrobacterium tumefaciens* strain *GV3101::pMP90* by electroporation. *A. thaliana* transformation was performed by the floral dip method with *A. tumefaciens* strains carrying the binary expression plasmids, and kanamycin-resistant $T_1$ plants were selected. $T_2$ lines that showed a 3:1 segregation ratio for kanamycin resistance were selected, as the segregation ratio supports the idea that these lines contain a single T-DNA insertion.

To examine inflorescence phenotypes, 16 plants each were grown from eight lines: Col-0, three $T_2$ lines transformed with the construct for *BLH9*, and four $T_2$ lines transformed with the construct for *AtBLH9*. To eliminate the effects of kanamycin selection on plant growth, all seeds of $T_2$ plants and Col-0 were grown in Murashige and Skoog medium without kanamycin. The expression levels of *BLH9* and *AtBLH9* in $T_2$ plants were confirmed by RT-qPCR (S2 Data). The primers used in RT-qPCR are listed in S1 Text. The $T_2$ plants were separated into four groups based on the relative expression levels of the inserted genes calculated by the $2^{-\Delta\Delta CT}$ method (S30 Fig): *BLH9* Control ($2^{-\Delta\Delta CT} \leq 2$), *BLH9 OX* ($2^{-\Delta\Delta CT} > 2$), *AtBLH9* Control ($2^{-\Delta\Delta CT} \leq 2$), and *AtBLH9 OX* ($2^{-\Delta\Delta CT} > 2$). At 30 days after the plants were transplanted to soil, three phenotypes were measured: inflorescence height, mean fruit length, and internode length. All phenotypic data are listed in S3, S4, and S5 Datas. The mean inflorescence heights and mean fruit lengths were compared by Wilcoxon rank sum test between the five groups: Col-0, *BLH9* Control, *BLH9 OX*, *AtBLH9* Control, and *AtBLH9 OX*. The variations in internode lengths between the five groups were compared by $F$-test. The $p$-values were adjusted by Bonferroni correction.

All detailed materials and methods are shown in S1 Text. All sequencing data are available under the Umbrella Bio-Project number PRJNA1223176. The genome assemblies and annotation files are available at zenodo: https://doi.org/10.5281/zenodo.14899024.

## Supporting information

**S1 Text. Supplementary materials and methods.** 1. Materials. 2. Reference assembly. 3. Generation of chromosomes using pseudo-testcross methods. 4. Identification of sex-linked regions by association analysis and mapping coverage analysis. 5. Genome structures of the X and Y chromosomes. 6. Identification of candidate genes for sex determination. 7. Identification of candidate miRNAs for sex determination. 8. Overexpression of *BLH9* and *AtBLH9* in *Arabidopsis thaliana*. (DOCX)

**S1 Fig. Morphological characteristics of *Dioscorea tokoro*.** (TIF)

**S2 Fig. Estimation of genome size of *Dioscorea tokoro* by flow cytometry.** (TIF)

**S3 Fig. Schematic diagram of the pseudo-testcross method.** (TIF)

**S4 Fig. F$_1$ progeny of *Dioscorea tokoro* used for linkage map construction and linkage analysis.** (TIF)

**S5 Fig. Schematic diagram for the selection of single-nucleotide polymorphism (SNP)-type and presence/absence polymorphism (PA)-type heterozygous markers from restriction site-associated DNA sequencing (RAD-seq) data.** (TIF)

**S6 Fig. Restriction site-associated DNA sequencing (RAD-seq)-based linkage map of the Waka1 (female) haplotype 1 reference genome generated by the pseudo-testcross method using 186 F$_1$ progeny.** (TIF)

**S7 Fig. Restriction site-associated DNA sequencing (RAD-seq)-based linkage map of the Waka1 (female) haplotype 2 reference genome generated by the pseudo-testcross method using 186 F$_1$ progeny.** (TIF)

**S8 Fig. Restriction site-associated DNA sequencing (RAD-seq)-based linkage map of the Kita1 (male) haplotype 1 reference genome generated by the pseudo-testcross method using 186 F$_1$ progeny.** (TIF)

**S9 Fig. Restriction site-associated DNA sequencing (RAD-seq)-based linkage map of the Kita1 (male) haplotype 2 reference genome generated by the pseudo-testcross method using 186 F$_1$ progeny.** (TIF)

**S10 Fig. Integrated genetic and physical maps of the Waka1 (female) haplotype 1 reference genome.** (TIF)

**S11 Fig. Integrated genetic and physical maps of the Waka1 (female) haplotype 2 reference genome.** (TIF)

**S12 Fig. Integrated genetic and physical maps of the Kita1 (male) haplotype 1 reference genome.** (TIF)

**S13 Fig. Integrated genetic and physical maps of the Kita1 (male) haplotype 2 reference genome.**
(TIF)

**S14 Fig. Schematic diagram of the association analysis of sex phenotypes and DNA markers.**
(TIF)

**S15 Fig. Manhattan plots of log-transformed *q*-values (–log$_{10}$(*q*)) values between markers and sex phenotype showing genomic regions associated with sex phenotype.**
(TIF)

**S16 Fig. Manhattan plot of log-transformed *q*-values (–log$_{10}$(*q*)) between markers and sex phenotype showing genomic region associated with sex phenotype when the same numbers of female and male F$_1$ progeny were used.**
(TIF)

**S17 Fig. Simple interval mapping (SIM) curves detecting associations between the genotypes and sex phenotypes.**
(TIF)

**S18 Fig. Alignment depths of male and female parents in all chromosomes of the Waka1 (female) haplotype 1assembly.**
(TIF)

**S19 Fig. Alignment depths of male and female parents in all chromosomes of the Kita1 (male) haplotype 1 assembly.**
(TIF)

**S20 Fig. Alignment depths on the X chromosome of male and female individuals from northern, central, and southern Japan.**
(TIF)

**S21 Fig. Alignment depths on the Y chromosome of male and female individuals from northern, central, and southern Japan.**
(TIF)

**S22 Fig. Alignment depths of male and female parents in chromosome 3 of the Waka1 (female) haplotype 2 and Kita1 (male) haplotype 2 assembly.**
(TIF)

**S23 Fig. Sex-associated regions suggested by modelling of the allele and genotype frequencies by SDpop [63].**
(TIF)

**S24 Fig. Sex-associated regions suggested by association analysis based on RADSex [64] with restriction site-associated DNA sequencing (RAD-Seq) of 127 F$_1$ flowering progeny.**
(TIF)

**S25 Fig. Estimation of synonymous site divergence (*d$_S$*) between Y-specific genes and their possible orthologous genes (gametologs) in the X-specific region.**
(TIF)

**S26 Fig. Distribution of synonymous site divergence (*d$_S$*) values of all gene pairs across X and Y chromosomes.**
(TIF)

**S27 Fig. Phylogenetic tree of protein sequences of BLH homologues from *D. tokoro* and TALE superfamily proteins from *A. thaliana*.**
(TIF)

**S28 Fig. Phylogenetic tree of cDNA sequences of *HSP90* homologues from *D. tokoro* and *AtHSP90* protein family from *A. thaliana*.**
(TIF)

**S29 Fig. *BLH9* and *HSP90* are male specific in natural populations.**
(TIF)

**S30 Fig. Relative expression levels of *BLH9* and *AtBLH9* in $T_2$ individuals harboring *35S::BLH9* and *35S::AtBLH9*.**
(TIF)

**S1 Table. Summary of the genome assemblies of Waka1 (female) and Kita1 (male).**
(XLSX)

**S2 Table. Summary of predicted genes in the genome assemblies of Waka1 (female) and Kita1 (male).**
(XLSX)

**S3 Table. List of genes located in X-specific region.**
(XLSX)

**S4 Table. Differential expression analysis of miRNAs located on Y chromosome.**
(XLSX)

**S5 Table. BLASTX analysis of the two candidate genes.**
(XLSX)

**S1 Data. Summary of RAD-seq data from Waka1, Kita1, and 186 $F_1$ plants.**
(XLSX)

**S2 Data. Ct values for all transformed $T_2$ plants confirmed by RT-qPCR.**
(XLSX)

**S3 Data. Main inflorescence lengths of Col-0 and $T_2$ plants.**
(XLSX)

**S4 Data. Fruit lengths of Col-0 and $T_2$ plants.**
(XLSX)

**S5 Data. Internode lengths of Col-0 and $T_2$ plants.**
(XLSX)

## Acknowledgments

We would like to thank members of the Crop Evolution Laboratory at Kyoto University and Hiroki Matsuo of the Plant Pathology Laboratory at Kyoto University for advice on genome analysis. We also thank Yui Niihara for the support with the experiment performed at Kyoto Sangyo University. We are grateful to Katsuyoshi Kubota and the staff of the Kasuya Research Forest and Kyushu University and the Kosaji villagers in Koka, Shiga Pref., for their cooperation with sample collection. We also thank Adam Bogdanove for invaluable advice and support. Genome analyses were conducted on the supercomputer at ACCMS, Kyoto University, and the NIG supercomputer at the ROIS National Institute of Genetics.

## Author contributions

**Conceptualization:** Aoi Kudoh, Ryohei Terauchi.

**Data curation:** Aoi Kudoh, Satoshi Natsume, Toshiyuki Sakai.

**Formal analysis:** Aoi Kudoh, Yu Sugihara, Hiroaki Kato, Akira Abe, Kaori Oikawa, Motoki Shimizu, Kazue Itoh, Mai Tsujimura, Yoshitaka Takano, Toshiyuki Sakai, Hiroaki Adachi, Atsushi Ohta, Mina Ohtsu, Toru Terachi, Ryohei Terauchi.

**Funding acquisition:** Ryohei Terauchi.

**Investigation:** Aoi Kudoh, Satoshi Natsume, Yu Sugihara, Hiroaki Kato, Akira Abe, Kaori Oikawa, Motoki Shimizu, Kazue Itoh, Mai Tsujimura, Yoshitaka Takano, Toshiyuki Sakai, Hiroaki Adachi, Atsushi Ohta, Mina Ohtsu, Takuma Ishizaki, Toru Terachi, Ryohei Terauchi.

**Methodology:** Aoi Kudoh, Hiroaki Kato, Yoshitaka Takano, Toshiyuki Sakai, Hiroaki Adachi, Atsushi Ohta, Takuma Ishizaki, Toru Terachi, Ryohei Terauchi.

**Project administration:** Yoshitaka Takano, Ryohei Terauchi.

**Resources:** Satoshi Natsume, Hiroaki Kato, Akira Abe, Kaori Oikawa, Motoki Shimizu, Kazue Itoh, Mai Tsujimura, Toshiyuki Sakai, Hiroaki Adachi, Mina Ohtsu, Takuma Ishizaki, Toru Terachi, Ryohei Terauchi.

**Software:** Satoshi Natsume, Yu Sugihara, Akira Abe, Toshiyuki Sakai.

**Supervision:** Yoshitaka Takano, Toshiyuki Sakai, Hiroaki Adachi, Toru Terachi, Hideki Innan, Ryohei Terauchi.

**Validation:** Aoi Kudoh, Motoki Shimizu, Hideki Innan, Ryohei Terauchi.

**Visualization:** Aoi Kudoh, Yu Sugihara, Toshiyuki Sakai, Mina Ohtsu.

**Writing – original draft:** Aoi Kudoh, Ryohei Terauchi.

**Writing – review & editing:** Aoi Kudoh, Hideki Innan, Ryohei Terauchi.

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
