## [Decision Letter · Decision Letter 0]

15 Jun 2025

PGENETICS-D-25-00572

Whole-genome sequencing reveals the molecular basis of sex determination in the dioecious wild yam Dioscorea tokoro

PLOS Genetics

Dear Dr. Terauchi,

Thank you for submitting your manuscript to PLOS Genetics. After careful consideration, we feel that it has merit but does not fully meet PLOS Genetics's publication criteria as it currently stands. Therefore, we invite you to submit a revised version of the manuscript that addresses the points raised during the review process.

Please submit your revised manuscript within 60 days Aug 14 2025 11:59PM. If you will need more time than this to complete your revisions, please reply to this message or contact the journal office at plosgenetics@plos.org. Please include the following items when submitting your revised manuscript:

We look forward to receiving your revised manuscript.

Kind regards,

Tatiana Giraud

Academic Editor

PLOS Genetics

Angela Hancock

Section Editor

PLOS Genetics

Aimée Dudley

Editor-in-Chief

PLOS Genetics

Anne Goriely

Editor-in-Chief

PLOS Genetics

**Additional Editor Comments :**

We have now received three referee reports for your manuscript on the elucidation of the genetics of sex determinism in the wild yam Dioscorea tokoro. All referees found merits in the manuscript; however, several serious concerns were raised, in particular about the assembly quality, the lack of clarity about SNP analyses, non-optimal mapping for RNAseq investigation, the lack of dS analyses, the lack of support for hemizygosity and allelic relationships, as well as the lack of discussion of the apparent gain-of-function mutations, of the relevant literature and of technical limits.

I also found that the study is very interesting, with the discovery of the sex chromosomes and of candidate genes for sex determination in a new plant, that are much less studied than animals on these aspects. The study uses both genomics and experimental validation. However, I also have to agree that the manuscript lacks clarity and important analyses, and that some results lack supports due do non-optimal methodological choices. In addition, the beginning of the introduction and the end of the discussion should broaden the context, by replacing the findings in comparison to other plant genera too. The context in the introduction and the discussion is sometimes awkwardly explained (see some examples below). More importantly, the inference of the sex determining region being in the pericentromeric region seems unwarranted: transposable elements also accumulate in non-recombining regions; the presence of many TEs is not sufficient evidence for being a centromeric region. The statement L214 also seems wrong about TEs suppressing recombination, the causality is opposite, with TEs accumulating on non-recombining regions. I also agree with the referees that some classical analyses are missing, such as plotting the synonymous divergence per gene between the X and the Y chromosome along the X chromosome to further support the location of the sex-determining region and to look for stepwise extension of recombination suppression. The results P12 also lack some features of the sex-determining region: size, number of genes, TE enrichment, …

I would encourage resubmission if you are able to revise the manuscript along these lines. The referees also provide a list of excellent additional suggestions and questions, which should also be addressed.

More specific comments:

-L45: the male or female sex is heterogametic, not the sex chromosome

-L47-48: unclear, explain better and cite examples; or delete the sentence, not sure this is really useful for the paper

-L48-49: the transition between the genetic system (type of sex chromosomes) and the genes involved is abrupt, as it if the two concepts were confounded, while they are completely different.

-L68-69: unclear if different systems have been proposed for the same species or different species.

-L77-78: awkward wording

-L89: a ZZ/ZW system

-L124-125: explain what is Waka1 and Kita1; how do you know this is a XX or XY genotype?

-L130: contig assemblies

-L138: explain what is a pseudo-testcross approach

-L173: a test or tests

-L180 a component

-L184: what are PA markers? avoid abbreviations as much as possible

-L233/ clarify what is the linkage distance exactly and how it has been measured.

-L235: the pattern

-L237: I do not understand, there seems to be several inversions

-L262-268: this should be listed in a Table rather than in the text

-LL342: this should be stated in the result section, with the number of genes

-L346, 355 and elsewhere: no abbreviation at the beginning of a sentence

-Do not cite Figures in the discussion, they should have been explained well enough in the result section.

-L366-367: unclear if you mean the genus or the species; the genus is cited but the introduction gives a lot of results on this across the genus?

-L393-L394: not sure this inference is warranted, it is not because you have two candidate genes that the cited hypothesis is supported: only one could actually act, or one could have been recruited later depending on the exact function.

-The discussion recapitulates too much the results without broadening the context.

-All abbreviations should be defined in all legends.

-Gypsy elements have been renamed into Ty3 elements

-The legend in Figure 3 includes unwarranted inference about pericentromeric regions (see above)

**Journal Requirements:**

At this stage, the following Authors/Authors require contributions: Aoi Kudoh, Satoshi Natsume, Yu Sugihara, Hiroaki Kato, Akira Abe, Kaori Oikawa, Motoki Shimizu, Kazue Itoh, Mai Tsujimura, Yoshitaka Takano, Toshiyuki Sakai, Hiroaki Adachi, Atsushi Ohta, Mina Ohtsu, Takuma Ishizaki, Toru Terachi, and Ryohei Terauchi. Please ensure that the full contributions of each author are acknowledged in the "Add/Edit/Remove Authors" section of our submission form.

The list of CRediT author contributions may be found here: https://journals.plos.org/plosgenetics/s/authorship#loc-author-contributions

- TM on page: 23.

Potential Copyright Issues:

i) Please confirm (a) that you are the photographer of 1(A-D), 4(A-D), 5(B, D, E), S1(A-M), and S3(A-C), or (b) provide written permission from the photographer to publish the photo(s) under our CC BY 4.0 license.

ii) Figures 1(E, and F) 2A. Please confirm whether you drew the images / clip-art within the figure panels by hand. If you did not draw the images, please provide (a) a link to the source of the images or icons and their license / terms of use; or (b) written permission from the copyright holder to publish the images or icons under our CC BY 4.0 license. Alternatively, you may replace the images with open source alternatives. See these open source resources you may use to replace images / clip-art:

iii) Figures S1N, S3D, S12, S13, and S16. Thank you for stating that the maps are created "with

Quantum Geographic Information System (QGIS) software version 3.16.0 (QGIS Development Team

2022). The base map was obtained from the Geospatial Information Authority of Japan, 2020." Please (a) provide a direct link to the base layer of the map (i.e., the country or region border shape) and ensure this is also included in the figure legend; and (b) provide a link to the terms of use / license information for the base layer image or shapefile. We cannot publish proprietary or copyrighted maps (e.g. Google Maps, Mapquest) and the terms of use for your map base layer must be compatible with our CC BY 4.0 license.

6) Thank you for indicating that the data is available in  (zenodo: 10.5281/zenodo.14899024). Please provide a direct link to access the dataset.

7) Thank you for stating "There is no competing interests among the coauthors." If you have no competing interests to declare, please state "The authors have declared that no competing interests exist".

8) We noted that all the files are uploaded to the online submission form with the description " Cover Letter." Please amend the description of the main file to "manuscript " and the supplementary files to "supplemental.

**Reviewers' comments:**

Reviewer's Responses to Questions

Reviewer #1: Kudoh et al. present in this manuscript a high-quality genome assembly of Dioscorea tokoro. Through a combination of comparative analyses between male and female populations, mapping of random sequencing reads, and direct comparison of the X and Y chromosomes, they identify the sex-determining gene as residing in the pericentromeric region of the Y chromosome. They further narrow down the candidate genes using detailed expression data and conduct functional analyses for some of the candidates.

First, the genome assembly, based on current standards including ONT data and extensive use of genetic anchoring, results in a highly refined genome. From this assembly, the authors successfully identified X- or Y-specific regions, while recombination inactivation would not substantially evolve surrounding sex-determining regions in this species. This suggests that the sex chromosomes are either evolutionarily recent or have remained homomorphic over time.

They performed transformation experiments on a BLH-like sex-determining candidate gene using Arabidopsis, a model plant, which is a sound decision. This is because, from a theoretical standpoint regarding the evolution of dioecy, whether under a one-factor or two-factor model, at least one of the sex-determining factors is likely to act as a dominant negative. In the evolutionary context, such a factor would function as a gain-of-function suppressor of feminization. From this perspective, selecting a model plant for functional validation is highly appropriate, and given prior studies of sex-determining genes, it is reasonable to expect that transformation in a model species will produce informative phenotypes, an expectation that has been met in several past cases, like in Diospyros or Actinidia.

The identification of candidate sex-determining factors in a plant species like Dioscorea, which has undergone multiple transitions in its sex determination system, is an outstanding work. The system identified in this study could serve as a starting point for gaining various new insights in sex chromosome/determinant turnovers.

Overall, I nicely enjoyed reading this manuscript. I highly rate the authors’ detailed analyses throughout the study. I would like to raise three major (but simple) points of concern here. While it may not be absolutely necessary to address all of them, I am confident that these comments would help improve the quality of this manuscript.

1. It would further improve the manuscript if some discussions are added regarding the evolutionary context and estimated age of the sex-determining region. This is particularly relevant given that Dioscorea species are known to exhibit frequent turnovers of the sex-determining locus (or potentially genes). How ancient this sex-determining factor (in D. tokoro) is, and over what evolutionary timescale it has persisted, even in the absence of a fully developed recombination-suppressed region, is a key question. To address this, the detection of synonymous substitution rates (dS) between X and Y alleles (or dSxy) would be informative. In the MSY (male-specific region of the Y chromosome), even if synteny or sequence conservation is no longer detectable, X–Y alleles often remain scattered throughout the region (please see an example in Silene latifolia, Moraga et al. 2025; Akagi et al. 2025). By analyzing the distribution of dS values among these alleles, one can infer the age and evolutionary trajectory of the sex-determining region. An important question is whether this region is truly male-hemizygous. As will be mentioned later, it would be helpful to clarify whether the term “Y-specific” refers to a region that is strictly hemizygous (present only on the Y) or if X–Y allelic pairs still exist. If such gametologs are present, their dS values might provide insights into the approximate age of the establishment sex-determining system.

2. Regarding the phenotypic evaluation of the candidate gene, the result showing a “fruitless” phenotype would suggest the possibility that it functions as a SuF (suppressor of feminization) (by the way, more detailed phenotyping of the transformants, especially in flower organs, may be resulted in better conclusions, I guess). If so, it is likely a gain-of-function of a dominant-negative function. This raises the question of how such a gain-of-function might have occurred within the monophyletic clade given in Figure 5A. Typically, gain-of-function mutations arise through neofunctionalization of duplicated genes or changes in expression patterns (like Y-CLV3 in S. latifolia, SOFF for Asparagus, Shy Girl for Actinidia, all SuF), albeit with a few exceptions. Is there any indication of such a scenario in this case? In other words, does the branch leading to Dioscorea BLH1 in Fig. 5A show any unique features that might point to such a change? In the phylogeny, the use of Arabidopsis thaliana alone as a comparative reference may limit the resolution. A more focused analysis restricted to narrow BLH-like genes, including those from other species (especially monocots), could provide a clearer picture of the gene's evolutionary history and possible functional divergence.

3. The allelic status (or X–Y gametolog relationship) should be clearly described, especially for the candidate genes (e.g., those in Table 3). The authors suggest that these genes are expressed at higher levels in males than in females. If that is the case, do they have corresponding X alleles (or X gametologs)? Alternatively, are they Y-specific at the genomic level, lacking X allele (I don’t mean the X-Y synteny)? This point also relates to the described query regarding the detection of dS between X and Y alleles (or dSxy). The term “Y-specific” may be better to more finely defined (depending on the situation), in terms of “synteny level”, “genomic level”, “gene level (I mean no existence of the X-allele, even in MSY)” or “expression level”.

Minor points

L. 49 Diospyros lotus is generally called Caucasian persimmon.

L. 58 There has been about 10 “suggested” sex determinant candidates but most of them have not been experimentally validated. Also from this viewpoint, physiological characterization of the candidate gene (in this study) is worth to publishing.

L. 77-78 A representative genus including both XY and ZW systems would be Populus or Salix, of which molecular mechanisms have been (at least partially) uncovered (although I will leave it up to the authors to decide whether to include that information or not.)

Tables 1 & 2 may be better suited as supplementary data, considering the recent standards in genome biology for plants.

Similarly, I would place Tables 4 and 5 in the supplementary materials. Contents of Table 4 are barely mentioned in the main text. Table 5 presents BLAST results for the candidate genes. If anything is essential in the main text, it would be a figure showing their phylogenetic relationships, that is, a phylogenetic tree (like given in Fig. 5A).

Reviewer #2: Review Comments:

Kudoh et al. report the identification of sex chromosomes in the dioecious yam Dioscorea tokoro, an endemic species. Using Oxford Nanopore (ONT)-based genome assemblies of both male and female individuals, they identify two candidate sex-determining genes through transcriptomic analysis. Notably, this is the first study to investigate sex determination in D. tokoro specifically, although sex determination has been previously studied in other species of the Dioscorea genus, including some recent genomic studies. Given the variability of sex determination mechanisms within Dioscorea, this study provides a valuable contribution.

Among the strengths of the work, one of the two male haplotypes appears to be of good quality based on standard assembly metrics (Table 1). The identification of sex-determining gene candidates, as presented in Figures 4 and 5, looks solid and convincing. The authors use a comprehensive RNA-seq dataset, including both reproductive and vegetative tissues, multiple developmental stages, and both poly(A) and small RNA libraries. Additionally, one of the candidate genes was experimentally tested in Arabidopsis, which strengthens this part of their work.

On the downside, the female genome assembly is of relatively poor quality, with a high number of contigs and a low N50. The second male haplotype is also relatively short. These limitations are reflected in discrepancies between the genetic and physical maps shown in Figures S7–S9. The absence of Hi-C data, particularly for the female assembly, likely contributes to these inconsistencies.

A key issue is that the quality of the genome assemblies is not adequately discussed in the manuscript. While Table 1 contains important assembly statistics, these are not discussed. Similarly, the mismatches between the genetic and physical maps in Figures S7–S9 are not addressed. Moreover, the fact that different assembly pipelines were used for the male and female genomes is neither discussed nor justified. These points should be clearly addressed in the manuscript.

An important analysis is the coverage-based detection of sex-linked regions presented in Figure 2. From the methods, it appears that unmasked genomes were used for this analysis, and the authors acknowledge that repetitive elements may introduce noise. A straightforward solution, widely adopted in sex chromosome studies, is to perform male/female coverage comparisons using repeat-masked genomes. This would reduce background noise and improve signal specificity.

The section on recombination on the sex chromosomes (p. 12–13, lines 231–240) is somewhat unclear. The authors seem to be identifying the non-recombining region (NRY) and pseudoautosomal regions (PARs), which are standard concepts in sex chromosome studies. Explicitly using this terminology would improve clarity. It is also unclear how Figures S14 and S15 relate to the recombination data presented in Figures S7–S9; this relationship should be clarified.

Other comments:

• p. 4, line 38: The claim that “Approximately 30,000 dioecious plant species have been identified, representing 5–10% of angiosperm species” appears to be incorrect. According to Renner (2014), already cited in the manuscript, the estimated number is closer to 15,000 species. Please revise accordingly.

• Methods & Supplementary Methods: Please indicate the ONT flow cell chemistry used (e.g., R9.4.1, R10.4.1, etc.). The sequencing error profile varies significantly between chemistries, and this information is important for interpreting the assembly quality. A brief discussion of the limitations of ONT, particularly for older chemistries, would also be valuable. The quality score of the consensus sequence is not mentionned.

• p. 12, lines 215–217: The statement that “X- and Y-specific regions were located in a genomic region with lower gene density and higher density of repetitive DNA sequences, represented by gypsy long terminal repeats (LTRs)” (Fig. 3A, B) should be revised. The gene/repeat density does not directly indicate recombination suppression, although it may correlate. The primary evidence for suppressed recombination is the recombination map (Figures S7–S9). Please rephrase accordingly here and in the Discussion (pp. 17–18, lines 349–351).

• p. 17, lines 341–342: “This chromosome is differentiated into X- and Y-specific regions spanning ~3.4 and ~2.5 kb, respectively.” This should almost certainly be Mb, not kb.

• pp. 19–20, lines 393–405: The discussion on the two-gene model of sex determination and gene number comparisons with other systems is currently superficial. A more nuanced discussion, grounded in recent reviews and studies across plants, would enhance the manuscript. Please ensure all relevant literature is cited.

Reviewer #3: In this manuscript, Kudoh et al assembled the genome of one male and one female of the dioecious plant Dioscorea tokoro, using Oxford Nanopore Technology (ONT) long read sequencing. The ONT contigs were scaffolded using a genetic map based on RAD-seq of a family. The authors use a SNP analysis to determine the species carries an XY system. Then they use a coverage analysis to locate the sex-linked regions. The male-specific region is located in chromosome 3 pericentromeric region. Both the X and the Y haplotypes are lowly recombining in males and females. Using RNA-seq data from various tissues and developmental stages, the authors propose two sex-determining genes (male-biased in their expression) that could be responsible for sex determination. I found the manuscript interesting and it provides a lot of new resources on this species. However, some control analyses are required to ensure the proper assembly of the X and Y specific regions. I also believe there is a problem with the reference used in the RNA-seq analysis, potentially biasing the identification of sex-specific expression. Also, traditional analyses of sex chromosomes are lacking that I would like to suggest the authors include in a revised version of their manuscript. I also would like to suggest that the authors tone down their affirmations in some places, as the sex-determining genes have not been validated yet.

Assemblies:

It is not clear to me whether the X and Y haplotypes were assembled properly. It is particularly difficult to assemble them due to high repeat content. What is more, both haplotypes don’t recombine, making the genetic map useless to scaffold the X and Y-specific region. The authors could rely on other technology such as BioNano or Hi-C to check the quality of their assembly. If funding is an issue, I would recommend at the very least to map the ONT reads onto the assemblies and check if the X and Y-specific regions are contiguous. In the case of the male genome assembly, mapping of ONT reads onto haplotype 1 and 2 should not lead to heterozygosity. If heterozygosity is detected, the Y haplotype might in fact be a mosaic between the X and the Y. Haplotype 2 of the male genome seems problematic because it has fewer genes and it is shorter. If haplotype 2 really is the X haplotype, it should resemble the female genome assembly, but this is not the case, suggesting there are issues with haplotype 2 assemblies. Lines 162-164 of the main text should be toned down in agreement with these limitations.

SNP analysis:

It took me a lot of time to figure out what PA markers were, how they differed from other markers and how the SNP analysis had been made. In the end I had to guess. A figure illustrating the SNPs and their segregation would greatly help the understanding of the method (it could be supplementary). The authors should explain better in the text how they are able to exclude a ZW system using their SNP analysis. Importantly, excluding a W chromosome cannot be done when mapping is made on male haplotype 1, therefore the female reference is also very important and should be stressed (it is currently in the Supp data). The association analysis between SNPs and sex provides results that are not precise (the entire chromosome 3 is sex-linked), suggesting a lack of statistical precision. I would suggest repeating the analysis with equal numbers of males and females, rather than unbalanced numbers which increase variance and can inflate p-values. In Figure S10B it is confusing to see that male alleles are detected with the female reference. I suspect this is due to the presence of X-hemizygous SNPs? I suggest the authors separate in their analyses the different SNP and PA marker types: X-linked, Y-linked and show them in different Figures. This would help to identify Y-specific regions and X-hemizygous regions. Alternatively, the authors could use programs developed with this aim such as SEX-DETector or SDpop.

Read coverage analysis:

I would recommend re-running the read coverage analysis after excluding the coverage information from repetitive regions. I am currently not convinced that the identified region is indeed sex-linked, because coverage information is not consistent with the expected 1X in males and 2Xs in females.

RNA-seq analysis and candidate genes:

It is not appropriate to estimate female expression levels from mapping on a Y haplotype. X and Y haplotypes have diverged and prevent mapping of X reads onto the Y reference and vice versa. It is not surprising that the authors find male-biased genes with this method, since female expression is not well measured. I am afraid this bias might have affected the choice of the two candidate genes explored by the authors, which is why I am not convinced about the candidates. It is a shame because a lot of work was done on testing one candidate with over-expression mutants in A. thaliana.

Could the authors specify if stage S0 of developing buds already have organs formed inside? Ideally to study sex-determination the buds should be sampled before organ primordia are formed. It should be noted that female flowers carry aborted stamens, suggesting that sex-determination in females might happen at a later stage. The gene responsible for stamen abortion should therefore be studied at a later stage. Carpel abortion seems very early in males though. This could be discussed in terms of the pathways followed to evolve dioecy in this plant group: monoecy of gynodioecy ? Please see Barrett (2002).

The authors should justify better in the text why they wanted to study sRNAs.

A candidate sex-determining gene does not have to be Y-specific, it can be an X/Y genes with a dosage difference between males and females. It should be clarified if the candidate genes have no X copy, if so, where did they come from? Were they translocated from an autosome onto the Y? Do they have autosomal homologs?

I am not convinced that BLH9 prevents fruit formation, its phenotype in A. thaliana rather suggests it may lead to sexual dimorphism (differences in plant heights for example), but the plants still have fruits although smaller.

The gene candidates should be discussed in light of their dominance level. The Charlesworths “2 genes model” makes very precise predictions about the recessivity and dominance of mutations that can lead to the formation of sex chromosomes. For X/Y sex chromosomes, the female sterility factor is expected to be dominant and Y linked, while male sterility should be recessive and X linked. Are the results in agreement with this model? Other alternative models involve just one gene, such as in Salicaceae.

For the DESeq2 analysis, were genes filtered based on their fold-change? Were lowly expressed genes removed?

Other suggested analysis:

It is important to annotate properly X-Y homologs (with permissive blasts to accommodate for X-Y divergence) and identify Y-specific genes and X-hemizygous genes. These genes should be represented in a Figure comparing the X and Y haplotypes with links between X/Y pairs. Their numbers is informative for sex chromosome degeneration level (in particular the proportion of X-hemizygous genes). The age of the sex chromosomes should be estimated with dS on X-Y gene pairs. Potential strata on the X should be annotated also with dS. These results should be confronted with the other species of the clade. I suggest the authors inspire themselves from recent papers in the field such as She et al. (2025). It would help to present a Figure of a phylogeny of other species where the sex chromosomes have been characterized. It is not sufficient to say that the non-recombining sex-specific region is on the same chromosome to say that the system is ancestral, gene content should be compared as well as X-Y dS values. It is possible than the same chromosome became the sex chromosome independently twice.

Other suggestions:

The title at first read suggests that the authors have identified the sex-determining genes of D. tokoro, which we discover is not the case when reading the manuscript. Please tone done the conclusion of your title. In the abstract “BLH9 is thought to suppress female organ development” is too strong a statement. The gene is male-specific in D. tokoro and does not make A. thaliana female sterile, so there is no base yet to think BLH9 suppresses female organ development, for now it is just a candidate. Again in the abstract “we describe the molecular basis of sex determination in the monocot Dioscorea tokoro”, “describe” is too strong, please rather use explore. In the abstract “Within the Y-specific region, we identified two candidate genes that are likely involved in sex determination” please replace identified with propose. In the abstract “These results shed light on the complex evolution of dioecy in plants” is somewhat vague and in disagreement with the lack of discussion of this topic in the manuscript. I would either suggest the authors remove this sentence from the abstract or indeed discuss this topic in the discussion.

Many key references are missing from the sex chromosome literature in this manuscript, I suggest the authors include at the very least the following references: (Charlesworth and Charlesworth 1978; Renner 2014)

More references can be found in this recent review of the literature: (Saunders and Muyle 2024) especially references regarding work by Stephen Wright on lowly recombining regions prior to the formation of sex chromosomes.

In the introduction the authors ask very general questions about sex chromosomes and sex determination but these questions are not really answered in the discussion. The introduction should be re-written to tackle the more specific aims of this study and the clade studied. Very interesting questions can be asked with this Dioscorea model about sex chromosome turnover for example. A better comparison to other Dioscorea species with a Figure showing a phylogeny would be welcome.

Please write sex-determining genes (not determination) throughout the text.

The flowcell used for ONT sequencing should be specified, as well as the basecalling model version. For DNA-seq, what was the fragment length, read length and sequencing mode (paired-end?)?

Line 541 what do you mean by aligned transcriptomes? What version of R/qtl was used (line 600)?

References

Barrett SCH. 2002. The evolution of plant sexual diversity. Nat. Rev. Genet. 3:274–284.

Charlesworth B, Charlesworth D. 1978. A Model for the Evolution of Dioecy and Gynodioecy. The American Naturalist 112:975–997.

Renner SS. 2014. The relative and absolute frequencies of angiosperm sexual systems: dioecy, monoecy, gynodioecy, and an updated online database. Am. J. Bot. 101:1588–1596.

Saunders PA, Muyle A. 2024. Sex chromosome evolution: hallmarks and question marks. Mol Biol Evol:msae218.

She H, Liu Z, Xu Z, Zhang H, Wu J, Wang X, Cheng F, Charlesworth D, Qian W. 2025. Genome sequence of the wild species, Spinacia tetrandra, including a phased sequence of the extensive sex-linked region, revealing partial degeneration in evolutionary strata with unusual properties. New Phytologist 246:2765–2781.

**Have all data underlying the figures and results presented in the manuscript been provided?**

Large-scale datasets should be made available via a public repository as described in the *PLOS Genetics*
data availability policy, and numerical data that underlies graphs or summary statistics should be provided in spreadsheet form as supporting information., and numerical data that underlies graphs or summary statistics should be provided in spreadsheet form as supporting information., and numerical data that underlies graphs or summary statistics should be provided in spreadsheet form as supporting information., and numerical data that underlies graphs or summary statistics should be provided in spreadsheet form as supporting information.

Reviewer #1: Yes

Reviewer #2: Yes

Reviewer #3: Yes

PLOS authors have the option to publish the peer review history of their article (what does this mean?). If published, this will include your full peer review and any attached files.). If published, this will include your full peer review and any attached files.). If published, this will include your full peer review and any attached files.). If published, this will include your full peer review and any attached files.

...

Reviewer #1: **Yes:** Takashi AkagiTakashi AkagiTakashi AkagiTakashi Akagi

Reviewer #2: No

Reviewer #3: No

**Figure resubmission:**
---

## [Decision Letter · Decision Letter 1]

22 Oct 2025

PGENETICS-D-25-00572R1

Whole-genome sequencing reveals a possible molecular basis of sex determination in the dioecious wild yam Dioscorea tokoro

PLOS Genetics

Dear Dr. Terauchi,

Thank you for submitting your manuscript to PLOS Genetics. After careful consideration, we feel that it has merit but does not fully meet PLOS Genetics's publication criteria as it currently stands. Therefore, we invite you to submit a revised version of the manuscript that addresses the points raised during the review process.

Please submit your revised manuscript within 60 days Dec 21 2025 11:59PM. If you will need more time than this to complete your revisions, please reply to this message or contact the journal office at plosgenetics@plos.org. Please include the following items when submitting your revised manuscript:

We look forward to receiving your revised manuscript.

Kind regards,

Tatiana Giraud

Academic Editor

PLOS Genetics

Angela Hancock

Section Editor

PLOS Genetics

Aimée Dudley

Editor-in-Chief

PLOS Genetics

Anne Goriely

Editor-in-Chief

PLOS Genetics

**Additional Editor Comments:**

We have now received two referee reports for your revised manuscript on the molecular basis of sex determination in the dioecious wild yam Dioscorea tokoro. One of the referees still had major concerns, in particular about the lack of delimitation of the sex-determining region and inconsistencies in the GWAS and dS analyses. They recommend additional analyses, notably high-quality recombination maps and plotting the distribution of sex-associated SNPs along the sex chromosome. I concur with these suggestions and add a few further recommendations below. I encourage resubmission if the manuscript is carefully revised along these lines and the other excellent suggestions made by the referee.

-L15: similar as

-L95, L111, L126 and elsewhere: “employs” sounds too anthropomorphic

-L203: coverage instead of dosage? Dosage is usually used for expression and there can be compensation

-L230-234: still a bit awkward. Recombination suppression is a key step in sex chromosome evolution. The accumulation of repetitive DNA sequences is a conspicuous feature of non-recombining regions, not only “primitive”, whatever this means. There are several examples in fungi of recombination suppression on mating-type chromosomes without inversions (e.g., https://academic.oup.com/mbe/article/38/6/2475/6130827;
https://www.nature.com/articles/s41467-018-04380-9;
https://pubmed.ncbi.nlm.nih.gov/29074958/)

-L268-283: S is subscript in dS; the idea actually of dS was to plot it along the X chromosome gene order to see if there were evolutionary strata, please add this analysis (e;g., https://www.science.org/doi/10.1126/science.adj7430). I don’t find the “Fig S22BC ».

-L427, 441, 442, and elsewhere : do not refer to figures in the discussion. There is no figure S27 right ?

**Reviewers' comments:**

Reviewer's Responses to Questions

**Comments to the Authors:**

Reviewer #1: The authors properly revised the manuscript satisfying my requests, and I have no more comments. I am looking forward the publication.

Reviewer #2: Despite the additional data and analyses provided in the revised manuscript, I believe that important information is still missing.

Based on the GWAS, it is clear that chromosome 3 harbors the SDR. However, the precise boundaries of the SDR and the PARs remain unclear. This uncertainty prevents a full and reliable characterization of the SDR. Without a clear definition of these boundaries, it is difficult to robustly assess the gene content of the X and Y haplotypes of the SDR, the extent of X–Y differentiation, or the degree of Y degeneration.

The coverage analysis presented is not entirely convincing and does not allow proper characterization of the X and Y haplotypes within the SDR. The observation that masking the genome does not improve the signal is surprising (figures in the rebuttal), although it may result from low X–Y divergence. In any case, this suggests that coverage analysis is not the most appropriate approach to characterize this SDR, especially since this method was originally designed for highly diverged animal systems.

The dS analysis of X/Y gametologs reveals very low values, comparable to those observed for random orthologs across the genome (new Figure S22). Furthermore, only a few gametolog pairs were identified, and they appear scattered across chromosome 3 — which is unexpected given the low overall X–Y differentiation.

The genetic maps are somewhat confusing. Comparing male and female recombination maps is typically informative, as it can reveal the non-recombining Y haplotype and the boundaries of the PARs. At present, several chromosomes show large discrepancies between maps built from male-heterozygous versus female-heterozygous SNPs, and these maps are not consistent between the male and female reference genomes (Figures S20, S21). It is not entirely clear whether male and female recombination data have been directly compared.

I strongly recommend including the distribution of sex-linked SNPs along chromosome 3. The authors should map the density of sex-associated SNPs and/or male-heterozygous+female-homozygous (presumably XY) SNPs along this chromosome. Applying published methods for such analyses (e.g., refs. 1, 2, or RAD-seq–specific approaches such as ref. 3) would be highly beneficial. Cross-comparison of these results with high-quality male and female recombination data would help delineate the SDR and PAR boundaries, thereby greatly strengthening all downstream analyses.

References

1. https://gitlab.in2p3.fr/sex-det-family/sex-detector-plusplus

2. Käfer J, Lartillot N, Marais GAB, Picard F. Detecting sex-linked genes using genotyped individuals sampled in natural populations. Genetics. 2021 Jun 24;218(2):iyab053. doi: 10.1093/genetics/iyab053. PMID: 33764439; PMCID: PMC8225351.

3. Wilson CA, High SK, McCluskey BM, Amores A, Yan YL, Titus TA, Anderson JL, Batzel P, Carvan MJ 3rd, Schartl M, Postlethwait JH. Wild sex in zebrafish: loss of the natural sex determinant in domesticated strains. Genetics. 2014 Nov;198(3):1291-308. doi: 10.1534/genetics.114.169284. Epub 2014 Sep 18. PMID: 25233988; PMCID: PMC4224167.

**Have all data underlying the figures and results presented in the manuscript been provided?**

Large-scale datasets should be made available via a public repository as described in the *PLOS Genetics*
data availability policy, and numerical data that underlies graphs or summary statistics should be provided in spreadsheet form as supporting information., and numerical data that underlies graphs or summary statistics should be provided in spreadsheet form as supporting information., and numerical data that underlies graphs or summary statistics should be provided in spreadsheet form as supporting information., and numerical data that underlies graphs or summary statistics should be provided in spreadsheet form as supporting information.

Reviewer #1: Yes

Reviewer #2: None

PLOS authors have the option to publish the peer review history of their article (what does this mean?). If published, this will include your full peer review and any attached files.). If published, this will include your full peer review and any attached files.). If published, this will include your full peer review and any attached files.). If published, this will include your full peer review and any attached files.

...

Reviewer #1: No

Reviewer #2: No

**Figure resubmission:**
---

## [Decision Letter · Decision Letter 2]

10 Dec 2025

PGENETICS-D-25-00572R2

Whole-genome sequencing reveals a possible molecular basis of sex determination in the dioecious wild yam Dioscorea tokoro

PLOS Genetics

Dear Dr. Terauchi,

Thank you for submitting your manuscript to PLOS Genetics. After careful consideration, we feel that it has merit but does not fully meet PLOS Genetics's publication criteria as it currently stands. Therefore, we invite you to submit a revised version of the manuscript that addresses the points raised during the review process.

Please submit your revised manuscript within by Jan 09 2026 11:59PM. If you will need more time than this to complete your revisions, please reply to this message or contact the journal office at plosgenetics@plos.org. Please include the following items when submitting your revised manuscript:

We look forward to receiving your revised manuscript.

Kind regards,

Tatiana Giraud

Academic Editor

PLOS Genetics

Angela Hancock

Section Editor

PLOS Genetics

Aimée Dudley

Editor-in-Chief

PLOS Genetics

Anne Goriely

Editor-in-Chief

PLOS Genetics

**Additional Editor Comments:**

I was pleased to see that this manuscript has been carefully revised, and the Referee also found the revisions overall satisfactory, they only had a few additional suggestions left, in particular on the SDpop analysis and the dS analyses. I also have additional suggestions below.

Please revise carefully all figure legends: all abbreviations should be defined in all legends, all information should be given on what is seen on the figures in all legends, and edit them for clarity. All figures should be understandable on their own without reading the text or the other figures.

-Figure S23: the legend is highly unclear and incomplete. For example: dS should be defined, dS had been computed between X and Y gametologs right? This is unclear from the figure legend, it reads as if it was between X or between Y alleles? The red and blue traits are not defined.

-L256-259: unclear and awkward. Remove “especially”, as it has no relationship with the previous sentence. Delete also “and thought to be involved in sex chromosome evolution”, as it sounds vague and unclear. In addition, it is a typical feature of non-recombining regions, not only plant sex chromosomes.

-L292, L816 and elsewhere: synonymous, not silent

-L270-271: this is more like discussion, and with also discussion of sex-determining regions being located in pericentromeric regions in plants (e.g. in Silene latifolia and Rumex).

-L293: “to infer the evolutionary context”: this is too vague and unclear.

-L294-298: this is unclear, explain what are the hypotheses tested, and even in the introduction already

-L312-315: this is also the case in many animal and fungal systems, and should be introduced in introduction rather than in the result section

-L318-319 and elsewhere: the lack of evolutionary does not mean that divergence is recent, but that there has been no successive extension with time. It is the value of the dS in the sex-determining region that indicates that it is recent. All this needs to be explained in the introduction.

-L459 and elsewhere: do not refer to figures in the discussion

-L480: the dS analysis (S in subscript across the whole manuscript)

-L482: comma after that

**Reviewers' comments:**

Reviewer's Responses to Questions

**Comments to the Authors:**

Reviewer #2: I thank the authors for their efforts in addressing my previous comments.

I still have two comments:

1. The SDpop analysis of chr3-Y nicely shows the SDR boundaries (Fig. S20). I was expecting to observe a similar pattern for chr3-X, which is not the case. As far as I understand, separate mappings were performed against the haploid female reference genome and against the male reference genome (haplotype 1 only). It appears that mapping to the female genome may be less efficient; providing the percentage of mapped reads for both references would help clarify this. If BWA parameters were identical for both mappings, the discrepancy may stem from differences in sequence quality, as the female genome was generated using R9.4.1 flow cells, which have a higher error rate. Could the authors indicate the Phred quality scores for both assemblies?

2. Figure S22 clearly shows that the SDR is younger than the divergence between D. tokoro and its relatives. However, given that the dS distributions are non-normal, I would recommend reporting median rather than mean values. The gametologs showing dS values > 1 are surprising — do they belong to multigene families? Could these be paralogs rather than true X/Y gametologs?

In Figure S23, I suggest displaying dS values only for the gametologs. For PAR genes, only allelic differences are expected. Showing dS for all homologous genes, including paralogous ones is confusing.

**Have all data underlying the figures and results presented in the manuscript been provided?**

Large-scale datasets should be made available via a public repository as described in the *PLOS Genetics*
data availability policy, and numerical data that underlies graphs or summary statistics should be provided in spreadsheet form as supporting information., and numerical data that underlies graphs or summary statistics should be provided in spreadsheet form as supporting information., and numerical data that underlies graphs or summary statistics should be provided in spreadsheet form as supporting information., and numerical data that underlies graphs or summary statistics should be provided in spreadsheet form as supporting information.

Reviewer #2: Yes

PLOS authors have the option to publish the peer review history of their article (what does this mean?). If published, this will include your full peer review and any attached files.). If published, this will include your full peer review and any attached files.). If published, this will include your full peer review and any attached files.). If published, this will include your full peer review and any attached files.

...

Reviewer #2: No

**Figure resubmission:**
---

## [Decision Letter · Decision Letter 3]

10 Feb 2026

PGENETICS-D-25-00572R3

Whole-genome sequencing reveals a possible molecular basis of sex determination in the dioecious wild yam Dioscorea tokoro

PLOS Genetics

Dear Dr. Terauchi,

Thank you for submitting your manuscript to PLOS Genetics. After careful consideration, we feel that it has merit but does not fully meet PLOS Genetics's publication criteria as it currently stands. Therefore, we invite you to submit a revised version of the manuscript that addresses the points raised during the review process.

Please submit your revised manuscript within by Mar 12 2026 11:59PM. If you will need more time than this to complete your revisions, please reply to this message or contact the journal office at plosgenetics@plos.org. Please include the following items when submitting your revised manuscript:

We look forward to receiving your revised manuscript.

Kind regards,

Tatiana Giraud

Academic Editor

PLOS Genetics

Angela Hancock

Section Editor

PLOS Genetics

Aimée Dudley

Editor-in-Chief

PLOS Genetics

Anne Goriely

Editor-in-Chief

PLOS Genetics

**Additional Editor Comments:**

This manuscript has been further improved, but the Referee still had important concerns about the SDpop analysis, with a clear pattern when using the Y chromosome as a reference but not when using the X chromosome, and with the dS analysis, with higher values in the pseudo-autosomal region than in the non-recombining region. There may be paralogs included and it may be helpful to seek expert advice for resolving these issues.

**Reviewers' comments:**

Reviewer's Responses to Questions

**Comments to the Authors:**

Reviewer #2: I thank the authors again for their efforts in addressing my comments. In particular, the sequencing and assembly of the female genome represent a very important addition to this work. Unfortunately, this does not resolve the puzzling pattern obtained with SDpop when using the female genome as a reference. Figure S25 is much improved, but Figure S26, which shows dS values along the 3X and 3Y chromosomes, remains difficult to interpret. dS reaches very high values, much higher than expected for two alleles in a pseudoautosomal regions (PARs). In addition, dS values in the SDR seem actually somewhat lower than those in the PARs, which is again unexpected. Moreover, the sliding-window analysis shows markedly different patterns for the SDR-3X and SDR-3Y, with a peak present only in SDR-3Y, which I find difficult to understand. This figure could likely be further improved.

**Have all data underlying the figures and results presented in the manuscript been provided?**

Large-scale datasets should be made available via a public repository as described in the *PLOS Genetics*
data availability policy, and numerical data that underlies graphs or summary statistics should be provided in spreadsheet form as supporting information., and numerical data that underlies graphs or summary statistics should be provided in spreadsheet form as supporting information., and numerical data that underlies graphs or summary statistics should be provided in spreadsheet form as supporting information., and numerical data that underlies graphs or summary statistics should be provided in spreadsheet form as supporting information.

Reviewer #2: Yes

PLOS authors have the option to publish the peer review history of their article (what does this mean?). If published, this will include your full peer review and any attached files.). If published, this will include your full peer review and any attached files.). If published, this will include your full peer review and any attached files.). If published, this will include your full peer review and any attached files.

...

Reviewer #2: No

**Figure resubmission:**
---

## [Decision Letter · Decision Letter 4]

30 Mar 2026

PGENETICS-D-25-00572R4

Whole-genome sequencing reveals a possible molecular basis of sex determination in the dioecious wild yam Dioscorea tokoro

PLOS Genetics

Dear Dr. Terauchi,

Thank you for submitting your manuscript to PLOS Genetics. After careful consideration, we feel that it has merit but does not fully meet PLOS Genetics's publication criteria as it currently stands. Therefore, we invite you to submit a revised version of the manuscript that addresses the points raised during the review process.

Please submit your revised manuscript within by Apr 29 2026 11:59PM. If you will need more time than this to complete your revisions, please reply to this message or contact the journal office at plosgenetics@plos.org. Please include the following items when submitting your revised manuscript:

We look forward to receiving your revised manuscript.

Kind regards,

Tatiana Giraud

Academic Editor

PLOS Genetics

Angela Hancock

Section Editor

PLOS Genetics

Aimée Dudley

Editor-in-Chief

PLOS Genetics

Anne Goriely

Editor-in-Chief

PLOS Genetics

**Additional Editor Comments:**

I was pleased to see that this manuscript has been carefully revised. The referee still has some minor suggestions, but other than that, I congratulate the authors for this nice piece of work that will be a great contribution to the journal.

**Reviewers' comments:**

Reviewer's Responses to Questions

Reviewer #3: The manuscript has significantly improved since I last read it after its first submission. I have some last comments to further improve it before publication.

Regarding the dS analysis: please state well in the main text that you filtered dS values to be lower than 0.5.

In Figure 2F we can see some female reads mapping on the male haplotype 1, unlike stated in the text lines 237-238 that coverage is zero in females.

Line 258-260: the SDpop results suggest that both X and Y reads are able to map on male haplotype 1 and female haplotype 1, because X/Y SNPs are found. This result suggests that X and Y sequences are poorly diverged (which is consistent with the dS analysis).

However, in Figure 2E-F you show that mapping is 0.5 for male illumina reads on male haplotype 1. Which is contradictory with the SDpop analysis.

It is contradictory to have at the same time half coverage on the X in males and X/Y SNPs. It is usually one or the other. These correspond to different stages of sex chromosome evolution. Please refer to reviews on identifying sex chromosomes such as Palmer et al 2019 or Muyle et al 2017.

The only explanation I see is maybe how mapping was done. Did you map illumina reads on haplotypes one by one for coverage analysis and SDpop? Or did you combine all haplotypes into a single reference and map illumina reads on this combined reference for the coverage analysis? For SDpop, the only way to obtain X/Y SNPs was to map reads on a single haplotype. The procedure should be clarified in the Results and Methods.

Figure 2D should have another version zoomed on the X and Y specific region. This would be a Figure similar to Figure S25C, with ribbons connecting X/Y gametologs, please make it a main Figure, but zoom on the X/Y specific region and comment it in the main text. This Figure highlights rearrangements between the X and the Y and is important.

Lines 320-326 describing gene orthology should appear before the dS description line 313.

I believe Figure S27 should be a main Figure. It is a lot more interesting than Figure 3 for example.

The result showing synteny between D tokoro and alata should be discussed further. In Figure S27B it is not possible to say if the tokoro SDR is syntenic to the alata SDR, a Figure with a zoom would be needed. If the two SDRs are syntenic, it would be interesting to consider if they evolved in the ancestor of the two species, or if they result from convergent evolution. The dS analysis comes in handy to suggest it is convergent evolution.

Could you add the BLH9 and HSP90 gene names in Table 1 please to identify more easily the lines corresponding to these candidates.

Line 501 what do you mean by "the genes involved in sex determination may have diverged"?

I rather think the conclusion is that the same region became sex-linked convergently in the two species.

**Have all data underlying the figures and results presented in the manuscript been provided?**

Large-scale datasets should be made available via a public repository as described in the *PLOS Genetics*
data availability policy, and numerical data that underlies graphs or summary statistics should be provided in spreadsheet form as supporting information., and numerical data that underlies graphs or summary statistics should be provided in spreadsheet form as supporting information., and numerical data that underlies graphs or summary statistics should be provided in spreadsheet form as supporting information., and numerical data that underlies graphs or summary statistics should be provided in spreadsheet form as supporting information.

Reviewer #3: Yes

PLOS authors have the option to publish the peer review history of their article (what does this mean?). If published, this will include your full peer review and any attached files.). If published, this will include your full peer review and any attached files.). If published, this will include your full peer review and any attached files.). If published, this will include your full peer review and any attached files.

...

Reviewer #3: No

**Figure resubmission:**
---

## [Editor Report · Decision Letter 5]

7 Apr 2026

Dear Dr Terauchi,

We are pleased to inform you that your manuscript entitled "Whole-genome sequencing reveals a possible molecular basis of sex determination in the dioecious wild yam Dioscorea tokoro" has been editorially accepted for publication in PLOS Genetics. Congratulations!

Yours sincerely,

Tatiana Giraud

Academic Editor

PLOS Genetics

Angela Hancock

Section Editor

PLOS Genetics

Aimée Dudley

Editor-in-Chief

PLOS Genetics

Anne Goriely

Editor-in-Chief

PLOS Genetics

BlueSky: @plos.bsky.social

Comments from the reviewers (if applicable):

The authors have carefully revised the manuscript, and I have no concern left, I congratulate them for the nice study.

**Data Deposition**

If you have submitted a Research Article or Front Matter that has associated data that are not suitable for deposition in a subject-specific public repository (such as GenBank or ArrayExpress), one way to make that data available is to deposit it in the Dryad Digital Repository. As you may recall, we ask all authors to agree to make data available; this is one way to achieve that. A full list of recommended repositories can be found on our . As you may recall, we ask all authors to agree to make data available; this is one way to achieve that. A full list of recommended repositories can be found on our . As you may recall, we ask all authors to agree to make data available; this is one way to achieve that. A full list of recommended repositories can be found on our . As you may recall, we ask all authors to agree to make data available; this is one way to achieve that. A full list of recommended repositories can be found on our website....

http://datadryad.org/submit?journalID=pgenetics&manu=PGENETICS-D-25-00572R5

Additionally, please be aware that our data availability policy requires that all numerical data underlying display items are included with the submission, and you will need to provide this before we can formally accept your manuscript, if not already present. requires that all numerical data underlying display items are included with the submission, and you will need to provide this before we can formally accept your manuscript, if not already present. requires that all numerical data underlying display items are included with the submission, and you will need to provide this before we can formally accept your manuscript, if not already present. requires that all numerical data underlying display items are included with the submission, and you will need to provide this before we can formally accept your manuscript, if not already present.

**Press Queries**

If you or your institution will be preparing press materials for this manuscript, or if you need to know your paper's publication date for media purposes, please inform the journal staff as soon as possible so that your submission can be scheduled accordingly. Your manuscript will remain under a strict press embargo until the publication date and time. This means an early version of your manuscript will not be published ahead of your final version. PLOS Genetics may also choose to issue a press release for your article. If there's anything the journal should know or you'd like more information, please get in touch via plosgenetics@plos.org....

---

## [Editor Report · Acceptance letter]

PGENETICS-D-25-00572R5

Whole-genome sequencing reveals a possible molecular basis of sex determination in the dioecious wild yam Dioscorea tokoro

Dear Dr Terauchi,

We are pleased to inform you that your manuscript entitled "Whole-genome sequencing reveals a possible molecular basis of sex determination in the dioecious wild yam Dioscorea tokoro" has been formally accepted for publication in PLOS Genetics! Your manuscript is now with our production department and you will be notified of the publication date in due course.

With kind regards,

Anita Estes

PLOS Genetics

On behalf of:
